# ReST: Reshaping Token-Level Policy Gradients for Tool-Use Large Language Models

**Zihan Lin**[1,2,3*†], **Xiaohan Wang**[2*], **Jie Cao**[1], **Jiajun Chai**[2],
**Guojun Yin**[2‡], **Wei Lin**[2], **Ran He**[1‡]

[1]MAIS&NLPR, Institute of Automation, Chinese Academy of Sciences
[2]Meituan
[3]School of Advanced Interdisciplinary Sciences, University of Chinese Academy of Sciences

## Abstract

Large language models (LLMs) transcend passive generation and act as goal-directed agents by invoking external tools. Reinforcement learning (RL) offers a principled framework for optimizing these emergent tool-use policies, yet the prevailing paradigm relies exclusively on sparse outcome rewards and lacks consideration of the particularity of tool-use tasks, inflating policy-gradient variance and resulting in inefficient training. To better understand and address these challenges, we first establish a theoretical link between policy entropy and training stability of tool-use tasks, which reveals that structured, low-entropy tokens are primary determinants of rewards. Motivated by this insight, we propose **Res**haped **T**oken-level policy gradients (**ReST**) for tool-use tasks. ReST reshapes the policy gradient through entropy-informed token reweighting, progressively upweighting reasoning tokens as training proceeds. This scheme enables a smooth shift from structural correctness to semantic reasoning and stabilizes convergence in multi-turn tool-use tasks. Evaluation on BFCL and API-Bank shows that ReST outperforms other strong baselines, outperforming prior methods by up to $8.76\%$. When fine-tuned on a 4B base LLM, ReST further surpasses GPT-4o by $4.11\%$ on single-turn tasks and $1.50\%$ on multi-turn base tasks. Code is available at `https://github.com/1229095296/ReST_Tool_use_LLM`.

## 1 Introduction

The recent emergence of large language models (LLMs) has precipitated a paradigm shift from passive, data-driven models to proactive, goal-oriented autonomous agents with complex tool-use abilities (Cheng et al., 2024; Xi et al., 2025). Consequently, considerable research has focused on post-training LLMs to enhance their tool-use capabilities for domain-specific applications such as deep search and travel assistance. (Wang et al., 2025a; Feng et al., 2025a). Post-training for agentic LLMs centers on supervised fine-tuning (SFT) and Reinforcement learning (RL) with distinct trade-offs (Chu et al., 2025; Wang et al., 2025b). In tool-use tasks, RL demonstrates strong generalization and robust performance (Song et al., 2025; Qian et al., 2025; Feng et al., 2025b). Most approaches, however, assign a single outcome reward after multi-turn tool calls (Jin et al., 2025); despite gains in multi-hop search (Liu et al., 2025a), this paradigm introduces two challenges for general tool use: First, reward design is intrinsically noisy. Unlike multi-hop search with near-unique references, many practical tool-use tasks admit multiple valid outputs (e.g., recommendations). As a result, outcome-only rewards induce high-variance gradients and provide weak incentives for reasoning, even when augmented with LLM-as-a-judge or learned reward models (Wang et al., 2025c; Gu et al., 2025). Second, multi-turn tool-use tasks are systems-inefficient: per-sample interactions, concurrency control, and variable horizons depress throughput and inflate infrastructure cost, making tool-use RL substantially heavier than SFT and single-turn RL (Team et al., 2025a).

---

[*]Both authors contributed equally to this research.
[†]Work was done during an internship at Meituan.
[‡]Corresponding authors: `yinguojun02@meituan.com`, `ran.he@ia.ac.cn`

The challenges motivate a central question: Could a tool-use agent be trained by rewarding each turn with specific and low-variance rewards? Inspired by Deepseek-R1's use of GRPO with rule-based rewards, we compute rule-based matching scores between generated responses and ground truth. Recent work, such as ToolRL (Qian et al., 2025), has validated this approach, achieving promising results. Specifically, these studies decompose multi-turn tool calls into a series of single-turn tasks and establish rule-based matching rewards for efficient RL training. However, prior work often overlooks a crucial aspect: in tool-call settings, rule-based rewards are inherently concentrated on format tags, tool names, and key parameters, while reasoning and general-response tokens contribute little during the early stages of training. Uniformly treating all tokens dilutes the RL signal and is particularly detrimental for approaches lacking token-level critics (e.g., GRPO).

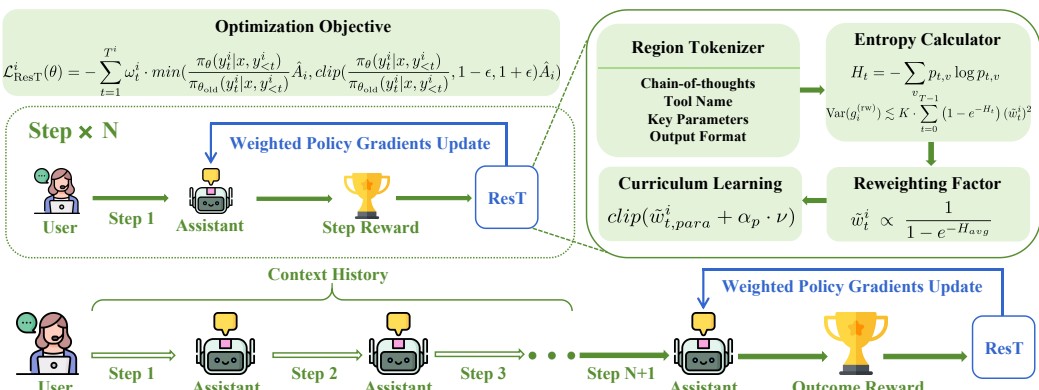

Figure 1: ResT decomposes multi-turn tool-use tasks into single-turn tasks and further reshapes the policy gradient according to the average entropy in different regions, enabling dense and effective reward signals.

To effectively and efficiently train tool-use agents, this paper proposes to **Res**hape **T**oken-level policy gradients based on single-turn RL training (**ResT**). As introduced in Figure 1, ResT factorizes multi-turn tool use into single-turn steps and reshapes policy gradients with region-level average entropy. This design emphasizes structured, low-entropy tokens such as tool names and parameters, yielding dense and effective reward signals. While RL for tool-use agents should ideally progress from format compliance to parameter accuracy and ultimately complex reasoning, such curricula are seldom adopted. In ResT, this progression is guided by adaptive weighting across token regions, providing a lightweight curriculum that further stabilizes training and achieves superior performance compared to strong baselines on established tool-use benchmarks. The contributions are summarized as follows:

1) We establish a theoretical and empirical link between policy entropy and training stability. Specifically, we demonstrate that a lower average entropy correlates with reduced variance in policy-gradient updates. This insight reveals that structured, low-entropy tokens—such as tool names and parameters—are primary determinants of the resulting rewards. By uncovering this connection, we provide a principled explanation for why emphasizing these tokens during training leads to more stable optimization and improved performance on tool-use tasks.

2) We introduce an entropy-aware, token-level reshaping mechanism that incorporates curriculum learning. Within this framework, tool-use and reasoning tokens are weighted by their region-level average entropy, which directly influences training stability and regulates how different token categories contribute to reward signals. As training progresses and entropy decreases, the weights for reasoning tokens are gradually increased. This curriculum-based approach stabilizes convergence and systematically enhances the model's reasoning capabilities in complex, multi-turn tool-calling scenarios.

3) Experimental results across multiple benchmark datasets show that our proposed method improves upon existing approaches by up to $8.76\%$, outperforming other strong baselines. Notably, when fine-tuned on the Qwen3-4B-2507 base model, our method outperforms GPT-4o (Hurst et al., 2024) by $1.50\%$ on multi-turn tool-use base tasks and $4.11\%$ on single-turn tool-use tasks. Further-

more, comprehensive ablation studies reveal that our curriculum-based reshaping outperforms static reward weighting by up to $4.86\%$.

## 2 RELATED WORK

### 2.1 REINFORCEMENT LEARNING FOR LLM REASONING

Research on reinforcement learning (RL) for large language models (LLMs) progressed from early paradigms such as reinforcement learning from human feedback (RLHF) (Kaufmann et al., 2024) and proximal policy optimization (PPO) (Schulman et al., 2017) to a class of methods based on preferences and policies, including direct preference optimization (DPO) (Rafailov et al., 2023), SimPO (Meng et al., 2024), and group relative policy optimization (GRPO) (Shao et al., 2024). Subsequent variants, such as dynamic sampling policy optimization (DAPO) (Yu et al., 2025) and value-augmented PPO (VAPO) (Yue et al., 2025), emphasize training stability and sample efficiency. Taken together, these advances indicate that RL can substantively enhance LLM reasoning, moving beyond surface-level pattern matching toward competencies such as logical inference and strategic planning (DeepSeek-AI et al., 2025; Team et al., 2025b).

Beyond sequence-level feedback, a line of work studies token-level feedback and credit assignment for fine-grained RLHF. TLCR (Yoon et al., 2024) uses GPT-4 to generate token-level preference labels and trains a discriminator to output continuous token rewards, thereby providing dense supervision at each position. TLDR (Fu et al.) constructs token-level reward models for multi-modal LLMs to detect hallucinations and support self-correction, highlighting the interpretability and practicality of token-wise rewards. Discriminative policy optimization with token-level Q-RMs (Chen et al., 2025) learns token-level Q-function–style reward models from preference data and applies them to RLHF scenarios such as mathematical reasoning. These methods explicitly construct token-level rewards or credit and then optimize policies under dense token-wise signals.

Curriculum RL provides another perspective on structuring learning signals. Classical work surveys curriculum design in RL and categorizes how task difficulty can be staged over time (Narvekar et al., 2020). Optimal-transport-based curriculum RL (Huang et al., 2022) formalizes curricula as Wasserstein geodesics between a source and target task distribution, generating intermediate tasks via gradual domain adaptation and analyzing the benefits of multi-stage task transitions. More recently, E2H Reasoner (Parashar et al., 2025) proposes an easy-to-hard reasoning curriculum for LLMs, and theoretically and empirically studies how combining RL with curricula improves sample efficiency on complex reasoning tasks. These approaches operate primarily at the task, environment, or prompt level by organizing training samples from easy to hard.

### 2.2 TOOL-USE AGENTS

Tool use extends LLMs from passive text generators to interactive systems capable of invoking external tools and APIs (Zhang et al., 2025; Hao et al., 2025). Existing approaches span prompt engineering and supervised fine-tuning (SFT) (Chen et al., 2023) on curated corpora (Qin et al., 2025; Hao et al., 2025; Zhang et al., 2024), alongside specialized architectures and retrieval-augmented pipelines that select tools conditioned on the query (Nguyen et al., 2024). Recent agentic frameworks further support multi-step planning and autonomous tool orchestration (Team et al., 2025a). However, most methods remain predominantly supervised, limiting exploration of diverse reasoning strategies and generalization to novel tool-use scenarios (Liu et al., 2025b).

To mitigate these limitations, RL formulations have been explored for tool-using agents. ToolRL (Qian et al., 2025) systematically compares SFT with GRPO (Shao et al., 2024); GIGPO (Feng et al., 2025b) adapts GRPO to multi-step tool use via multi-trajectory sampling and a two-level reward scheme; PVPO (Feng et al., 2025c) further reduces rollout dependence and alleviates local optima through a reference–anchor mechanism and data pre-sampling; meanwhile, Agent Lightning (Luo et al., 2025) decouples training from execution and employs hierarchical RL to convert arbitrary agent trajectories into transition-level learning signals. Despite these advances, practical RL methods still largely optimize at the sequence level, where coarse-grained rewards induce high variance in policy gradients, constraining reward granularity and impeding efficient credit assignment and optimization.

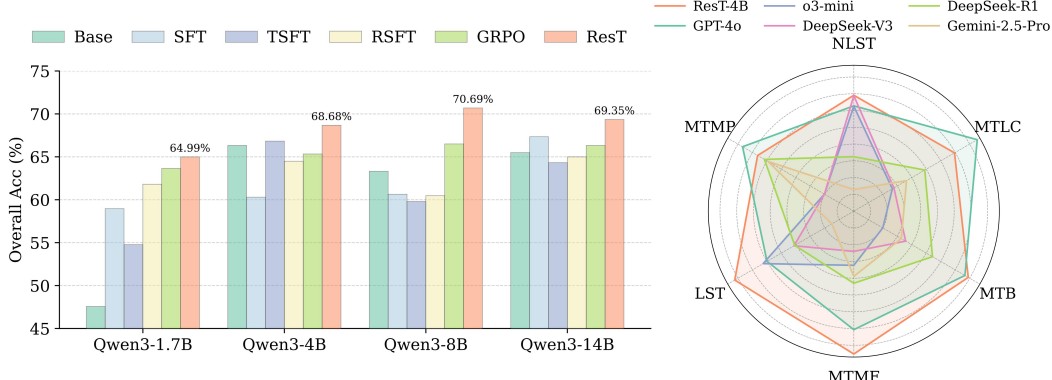

Figure 2: **Left:** Overall accuracy on the API-Bank test set. **Right:** Overall accuracy on the BFCL test set. Axes are: NLST: None-Live Single Turn, MTLC: Multi-Turn Long Context, MTB: Multi-Turn Base, MTMF: Multi-Turn w/ Missing Functions, LST: Live Single Turn, MTMP: Multi-Turn w/ Missing Parameters.

## 3 METHOD

In this section, we make two contributions. First, we develop a formal variance analysis that links token-level policy entropy to the stability of policy-gradient updates, and show that reward mass in tool-use tasks concentrates on structured, low-entropy tokens such as tool names, arguments, and format tags. Second, we introduce ResT, an entropy-aware token-level gradient reshaping scheme with a lightweight curriculum that initially upweights these reward-critical structural tokens and gradually shifts emphasis toward higher-entropy reasoning tokens, thereby reducing gradient variance, improving sample efficiency, and stabilizing training on complex tool-use tasks.

### 3.1 THEORETICAL MOTIVATION

Effective tool use often hinges on correctly generating just a handful of key tokens—those governing tool names, arguments, and output formats. Standard uniform reward schemes dilute these sparse yet decisive signals, inducing high-variance gradients and hampering stable convergence. To mitigate this, we develop a theoretical framework for reducing policy-gradient variance through entropy-aware token reweighting. Our analysis decomposes the gradient variance, establishes its connection to token-level entropy, and derives an optimal reweighting scheme that simultaneously enhances training efficiency and promotes stable convergence on high-precision tool-use tasks.

**Lemma 1** (Policy Gradient Variance Decomposition). *Let $J(\theta) = \mathbb{E}_{\tau \sim \pi_\theta}[R(\tau)]$ denote the expected return. For a single trajectory $\tau_i = (y_{i,0:T-1})$, define the trajectory-level gradient*

$$g_i = \left( \sum_{t=0}^{T-1} \nabla_\theta \log \pi_\theta(y_{i,t} \mid x, y_{i,<t}) \right) \hat{A}_i, \tag{1}$$

*where $\hat{A}_i$ is advantage function. Given $G$ i.i.d. trajectories $\{\tau_i\}_{i=1}^G$ from $\pi_\theta$, the mini-batch estimator is $\widehat{\nabla J} = (1/G) \sum_{i=1}^G g_i$. Then for each coordinate $k$ the variance satisfies*

$$\mathrm{Var}(\widehat{\nabla J}^{(k)}) = \frac{1}{G} \mathrm{Var}(g_i^{(k)}), \tag{2}$$

*If $\hat{A}_i = (r_i - \mu)/\sigma$ with $\mu = \mathbb{E}[r]$ and $\sigma^2 = \mathrm{Var}(r)$, then $\mathbb{E}[g_i] = \nabla_\theta J(\theta)$ and $\widehat{\nabla J}$ is unbiased. Proof is deferred to Appendix B.1.*

Lemma 1 establishes the variance scaling law under mini-batch sampling. Building on this decomposition, we next characterize the second-order moment of $g_i$ and connect it to token-level entropy.

**Lemma 2** (Second-Order Moment and Entropy Connection). *For the single-trajectory estimator $g_i$, the variance satisfies*

$$\mathrm{Var}(g_i) \leq \mathbb{E}\left[ \hat{A}_i^2 \left\| \sum_{t=0}^{T-1} J_t^\top s_t \right\|^2 \right], \tag{3}$$

where $s_t = e_{y_t} - p_t$ is the logit-space score, $p_t = \text{softmax}(z_t)$ is the token distribution with $z_t$ being the logits, and $J_t = \partial z_t / \partial \theta$ is the Jacobian. Moreover, the per-step score satisfies

$$\mathbb{E}[\|s_t\|^2] = 1 - \sum_v p_{t,v}^2 \leq 1 - e^{-H_t}, \tag{4}$$

where $H_t = -\sum_v p_{t,v} \log p_{t,v}$ is the Shannon entropy, with $v$ indexing over the token vocabulary. Proof is deferred to Appendix B.2.

Lemma 2 shows that variance fundamentally depends on both squared advantage and token entropy. This result provides a natural handle for reweighting gradient contributions across timesteps. For a sequence of length $T$, we denote the reweighted policy gradient estimator as:

$$g_i^{(\text{rw})} := \left( \sum_{t=0}^{T-1} \tilde{w}_t \nabla_\theta \log \pi_\theta(y_{i,t} \mid x, y_{i,<t}) \right) \hat{A}_i, \qquad \text{subject to} \quad \sum_{t=0}^{T-1} \tilde{w}_t = T, \tag{5}$$

**Theorem 1** (Variance Upper Bound for Entropy-Aware Reweighting). *The variance of the reweighted estimator $g_i^{(\text{rw})}$ is upper-bounded using Lemma 2:*

$$\text{Var}\big(g_i^{(\text{rw})}\big) \leq \mathbb{E}[\hat{A}_i^2] \sum_{t=0}^{T-1} \beta_t \, (\tilde{w}_t)^2, \tag{6}$$

where $\beta_t = \mathbb{E}[\|J_t\|_F^2 (1 - e^{-H_t})]$ quantifies the variance contribution of step $t$ by combining the Jacobian norm and the token entropy.

Building on this upper bound, we next seek the optimal reweighting scheme that minimizes the variance. This leads to a closed-form characterization of the weights.

**Theorem 2** (Optimal Entropy-Aware Reweighting). *Minimizing the variance bound in Theorem 1 with respect to the weights $\tilde{w}_t$ yields a closed-form solution: (Proof is deferred to Appendix B.3.)*

$$\tilde{w}_t^\star = \frac{T}{\sum_{u=0}^{T-1} \beta_u^{-1}} \cdot \frac{1}{\beta_t}, \tag{7}$$

which effectively down-weights sequence positions with larger intrinsic variance contributions. Substituting the optimal weights $\tilde{w}_t^\star$ into Theorem 1 gives the minimized variance bound:

$$\text{Var}\big(g_i^{(\text{rw})}\big) \leq \mathbb{E}[\hat{A}_i^2] \cdot \frac{T^2}{\sum_{t=0}^{T-1} \beta_t^{-1}}. \tag{8}$$

In practice, computing the exact value of $\beta_t$ depends on several variance terms that are difficult to estimate reliably at scale. Our variance decomposition shows that the leading structure of these terms is governed by token-level entropy. We therefore use an entropy-based surrogate to approximate $\beta_t$, which yields simple and scalable reweighting rules such as

$$\tilde{w}_t \propto \frac{1}{1 - e^{-H_{avg}}} \quad \text{or} \quad \tilde{w}_t \propto \frac{1}{H_{avg}}. \tag{9}$$

$H_{\text{avg}}$ denotes the average token-level entropy computed separately for each region in tool-use trajectories (format tags, tool names, key parameters, and chain-of-thought tokens). These region-wise entropy summaries preserve the qualitative inverse relationship between entropy and variance contribution implied by our analysis, and therefore provide efficient, entropy-based surrogates for the optimal weights in Equation 7.

These findings motivate a principled variance-reduction scheme for policy-gradient training on tool-use trajectories. In our method, open-ended tokens (e.g., chain-of-thought), which exhibit higher entropy (Besta et al., 2025) (see Figure D.1), are assigned lower weights than structured, low-entropy control tokens. This entropy-aware reweighting tightens variance bounds and improves sample efficiency while preserving the unbiasedness of the policy-gradient estimator.

### 3.2 ALGORITHM DESIGN

#### 3.2.1 REWARD SCORE

Rule-based reward mechanisms have been empirically validated to achieve robust performance across diverse settings. In line with prior work (Qian et al., 2025; Jin et al., 2025; Liao et al., 2024; Xie et al., 2025), our training objective incorporates components for both output format and execution correctness. Accordingly, we formulate the total reward as a weighted sum of a format matching score and a tool-calling correctness score. Format score of the $i$-th sample is determined by exact-match:

$$\mathcal{S}_{i,\text{format}} = \begin{cases} 1, & \text{if all required fields are complete in order} \\ 0, & \text{otherwise} \end{cases} \tag{10}$$

The tool-calling correctness score is based on three components: tool name, parameter names, and parameter values. We measure name accuracy using the Jaccard similarity coefficient, defined as the intersection over union of the predicted $P$ and ground-truth $G$ sets:

$$r_{\text{name}} = \frac{|N_G \cap N_P|}{|N_G \cup N_P|} \in [0,1], \quad r_{\text{para}} = \sum_{G_i \in G} \frac{|P_G \cap P_P|}{|P_G \cup P_P|} \in [0,|G|], \tag{11}$$

where $N_G$ and $N_P$ are the sets of ground-truth and predicted tool names, while $P_G$ and $P_P$ the corresponding sets of parameter names for a given tool. Due to the precision requirements of parameter values, we use an exact-match criterion for their evaluation:

$$r_{\text{value}} = \sum_{G_i \in G} \sum_{v \in \mathbf{v}(G_i)} \mathbf{1}[P_G[v] = P_P[v]] \in \left[0, \sum_{G_i \in G} |\mathbf{v}(G_i)|\right], \tag{12}$$

where $\mathbf{v}(\cdot)$ denotes the function that maps a graph $G_i$ to its corresponding parameter value set. The normalized correctness score is then computed as:

$$\mathcal{S}_{i,\text{acc}} = \frac{r_{\text{name}} + r_{\text{para}} + r_{\text{value}}}{1 + |G| + \sum_{G_i \in G} |\mathbf{v}(G_i)|}. \tag{13}$$

The final dynamic scaling reward $\mathcal{R}_{\text{final}}$ is then calculated as a weighted sum:

$$\mathcal{R}_{i,\text{final}} = \beta_a \mathcal{S}_{i,\text{acc}} * (1 - \bar{\nu}) + \beta_f \mathcal{S}_{i,\text{format}} * (1 - \bar{\nu}), \tag{14}$$

where $\beta$ denotes the relative importance and $\bar{\nu} \in (0,1)$ denotes the training progress.

#### 3.2.2 RESHAPING TOKEN-LEVEL POLICY GRADIENTS

ResT enables a more stable and effective use of rule-based rewards during model fine-tuning. Given a prompt containing user requirements and context, a policy $\pi_\theta$ generates $G$ multi-turn response sequences composed of plain text and tool invocation segments. For the $i$-th sample in a group $G$, the advantage function $\hat{A}_i$ are then calculated:

$$\hat{A}_i = \frac{\mathcal{R}_{i,\text{final}} - \frac{1}{G} \sum_{i=0}^{G-1} \mathcal{R}_{i,\text{final}}}{\sqrt{\frac{1}{G} \sum_{i=0}^{G-1} (\mathcal{R}_{i,\text{final}} - \frac{1}{G} \sum_{i=0}^{G-1} \mathcal{R}_{i,\text{final}})^2 + \delta}}, \tag{15}$$

where $\delta > 0$ is a numerical stability constant.

We first tokenize the response and then partition it into four weight sets: $\tilde{w}_{t,fmt}$ (format tags), $\tilde{w}_{t,name}$ (tool names), $\tilde{w}_{t,para}$ (tool parameters), and $\tilde{w}_{t,thk}$ (chain-of-thought). Reweighting factor $\tilde{w}_t$ is initialized according to the entropy-based reweighting equation 9.

Our curriculum dynamically adjusts weights based on progress $\nu$, enforcing a transition from mastering structural syntax to refining semantic strategy. While tool names retain constant high importance, we initially prioritize format tokens to ensure validity, subsequently shifting focus to semantic accuracy by annealing format weights while increasing parameter and chain-of-thought weights:

$$\tilde{w}_{t,fmt}(\nu) = \max(w_{\min}, \tilde{w}_{t,fmt} - \alpha_f \cdot \nu), \quad \tilde{w}_{t,para}(\nu) = \min(w_{\max}, \tilde{w}_{t,para} + \alpha_p \cdot \nu), \tag{16}$$

where $[w_{\min}, w_{\max}]$ denotes the clipping interval and $\alpha$ denotes the magnitude of the update. To reduce policy-gradient variance, encourage step-by-step reasoning, and improve parameter accuracy, the weights for chain-of-thought and tool parameter tokens are synchronized:

$$\tilde{w}_{t,thk}(\nu) = \min(w_{\max}, \tilde{w}_{t,thk} + \alpha_t \cdot \nu). \tag{17}$$

We obtain the normalized reweighting factor by its mean over the sequence:

$$\hat{w}_t := \max\left(w_{\min}, \min\left(\tilde{w}_t, w_{\max}\right)\right), \quad \bar{w} := \frac{1}{|T|} \sum_{t=1}^{T} \hat{w}_t, \quad w_t = \frac{\hat{w}_t}{\bar{w} + \delta}, \tag{18}$$

This normalization mechanism redistributes learning signals across tokens within each sample, governed by their relative policy entropy. By incorporating this reweighting factor with the advantage function, we derive the final optimization objective:

$$\mathcal{L}_{\mathrm{ResT}}(\theta) = -\frac{1}{G} \sum_{i=0}^{G-1} \sum_{t=0}^{T-1} \frac{\omega_t}{T} \cdot \min\left(r_{i,t}\hat{A}_i, \ \mathrm{clip}(r_{i,t}, 1-\epsilon, 1+\epsilon)\hat{A}_i\right), \tag{19}$$

where $r_{i,t} = \dfrac{\pi_\theta(y_{i,t} \mid x, y_{i,<t})}{\pi_{\theta_{\mathrm{old}}}(y_{i,t} \mid x, y_{i,<t})}$ and $\epsilon$ limits the magnitude of gradient updates. Our approach omits the KL loss term from the GRPO objective, which typically penalizes excessive policy deviations. In its place, ResT intricately modulates the exploration/exploitation trade-off via a combination of entropy-aware reweighting, clipping mechanisms, and curriculum learning. We validate this design choice through ablation studies in Figure A and Table A.1, which show that these components together yield stable and performant training dynamics even in the absence of an explicit KL penalty. The objective guides the policy toward structurally consistent and semantically accurate tool calls, while gradient reshaping alleviates variance in policy gradients, resulting in more stable and efficient alignment with task requirements. The complete algorithmic procedure is provided in Appendix C.

## 4 EXPERIMENTS

### 4.1 TRAINING DETAILS

We conducted comprehensive experiments using the verl 0.5.0 framework. We curate a mixed corpus to facilitate robust tool learning in RL: ToolACE (Liu et al., 2025b) targets the invoke-vs-answer decision in multi-step interactions; Hammer (masked) (Lin et al., 2025) randomizes tool and argument names to enforce description-grounded generalization; and XLAM (Zhang et al., 2024) supplies compositional tasks that require one or multiple tool calls per turn. Together, these sources supervise invocation timing, robust grounding, and compositional planning (Qian et al., 2025).

We convert multi-turn dialogue samples into single-step instances to provide denser and more fine-grained reward signals. Concretely, a dialogue with $K$ interaction steps is decomposed into $K$ sub-instances, each conditioned on the full preceding conversation as input and supervised with the model's current action (e.g., tool invocation or response) as the target. This decomposition strategy, adopted from the SWiRL framework (Goldie et al., 2025), substantially increases the density of training signals. Consequently, the model receives process-level supervision at each step, rather than only for the final outcome. This step-wise supervision fosters more stable learning of multi-step reasoning and tool use.

### 4.2 MAIN RESULTS

We evaluate six baselines spanning supervised and reinforcement-learning regimes: **Base Model**: instruction-tuned without task-specific fine-tuning. **SFT**: supervised fine-tuning on the task training set. **TSFT**: SFT with up-weighted loss (Huerta-Enochian & Ko, 2024) on tool-call tokens to prioritize invocation accuracy. **RSFT**: SFT with up-weighted loss on reasoning tokens to strengthen intermediate reasoning. **GRPO**: A critic-free reinforcement learning algorithm that leverages group-relative rewards to efficiently optimize tool-use policies. **SFT+GRPO**: initialize from an SFT checkpoint, then apply GRPO to isolate RL's marginal gains. **Dr.GRPO** (Liu et al., 2025c) removes

Table 1: **BFCL multi-turn results (updated June 14, 2025).** Metrics computed with official scripts. TSFT scales loss on tool-call tokens. RSFT scales loss on reasoning tokens. Best in **bold**.

| Model | Overall Acc | Base | Miss Func | Miss Param | Long Context |
|---|---|---|---|---|---|
| Qwen3-1.7B (Base) | 8.62% | 12.00% | 5.50% | 9.50% | 7.50% |
| Qwen3-1.7B (SFT) | 14.25% | 17.50% | 14.50% | 14.50% | 10.50% |
| Qwen3-1.7B (TSFT) | 13.50% | 18.00% | 14.50% | 11.50% | 10.00% |
| Qwen3-1.7B (RSFT) | 15.00% | 17.00% | 16.50% | 14.50% | 12.00% |
| Qwen3-1.7B (GRPO) | 14.50% | 17.00% | 15.50% | 12.50% | 13.00% |
| Qwen3-1.7B (SFT+GRPO) | 14.88% | 20.50% | 18.00% | 9.00% | 12.00% |
| Qwen3-1.7B (Dr.GRPO) | 15.25% | 16.50 % | 15.00% | 17.00% | 12.50% |
| Qwen3-1.7B (Ours, ResT) | **16.00%** | 20.00% | 16.00% | 15.00% | 13.00% |
| Qwen3-4B-2507 (Base) | 48.00% | 57.00% | 53.00% | 33.50% | 48.50% |
| Qwen3-4B-2507 (SFT) | 43.00% | 52.50% | 46.50% | 33.00% | 40.00% |
| Qwen3-4B-2507 (TSFT) | 48.25% | 58.00% | 52.00% | 32.50% | 50.50% |
| Qwen3-4B-2507 (RSFT) | 41.25% | 52.00% | 47.50% | 31.50% | 34.00% |
| Qwen3-4B-2507 (GRPO) | 41.62% | 48.50% | 45.50% | 27.00% | 45.50% |
| Qwen3-4B-2507 (SFT+GRPO) | 40.38% | 46.00% | 42.00% | 29.50% | 44.00% |
| Qwen3-4B-2507 (Dr.GRPO) | 48.62% | 58.00 % | 54.00% | 30.50% | 52.00% |
| Qwen3-4B-2507 (Ours, ResT) | **50.38%** | 62.50% | 56.00% | 33.00% | 50.00% |
| Qwen3-8B (Base) | 33.00% | 38.50% | 32.00% | 31.50% | 30.00% |
| Qwen3-8B (SFT) | 28.75% | 34.50% | 33.50% | 22.00% | 25.00% |
| Qwen3-8B (TSFT) | 29.00% | 32.50% | 34.00% | 24.00% | 25.50% |
| Qwen3-8B (RSFT) | 25.12% | 33.00% | 29.50% | 19.00% | 19.00% |
| Qwen3-8B (GRPO) | 36.00% | 42.00% | 42.00% | 28.50% | 31.50% |
| Qwen3-8B (SFT+GRPO) | 32.38% | 42.50% | 35.00% | 27.50% | 24.50% |
| Qwen3-8B (Dr.GRPO) | 38.12% | 39.50 % | 44.50% | 36.50% | 32.00% |
| Qwen3-8B (Ours, ResT) | **40.13%** | 50.50% | 45.00% | 32.00% | 33.00% |
| Qwen3-14B (Base) | 36.62% | 48.00% | 33.00% | 33.50% | 32.00% |
| Qwen3-14B (SFT) | 28.12% | 33.50% | 30.00% | 23.00% | 26.00% |
| Qwen3-14B (TSFT) | 27.62% | 34.50% | 34.00% | 17.50% | 24.50% |
| Qwen3-14B (RSFT) | 27.00% | 33.50% | 27.50% | 20.00% | 27.00% |
| Qwen3-14B (GRPO) | 38.88% | 50.00% | 43.50% | 26.50% | 35.50% |
| Qwen3-14B (SFT+GRPO) | 34.12% | 45.00% | 37.50% | 22.00% | 32.00% |
| Qwen3-14B (Dr.GRPO) | 37.00% | 46.00 % | 33.50% | 41.00% | 27.50% |
| Qwen3-14B (Ours, ResT) | **44.25%** | 57.00% | 45.00% | 36.50% | 38.50% |
| GPT-4o-2024-11-20 | 50.00% | 61.00% | 45.50% | 35.50% | 58.00% |
| GPT-4.5-Preview-2025-02-27 | 45.38% | 58.00% | 37.50% | 43.00% | 43.00% |
| GPT-4.1-2025-04-14 | 40.50% | 48.00% | 34.00% | 35.00% | 45.00% |
| o3-mini-2025-01-31 | 28.75% | 32.00% | 26.00% | 23.00% | 34.00% |
| Grok-3-beta | 45.50% | 54.00% | 39.00% | 42.00% | 47.00% |
| DeepSeek-V3-0324 | 29.87% | 41.00% | 21.00% | 23.00% | 34.50% |
| DeepSeek-R1-0528 | 38.88% | 49.50% | 31.50% | 32.00% | 42.50% |
| Gemini-2.5-Pro-Preview-05-06 | 34.62% | 39.50% | 29.50% | 31.50% | 38.00% |

standard-deviation normalization in the group-relative advantage and instead uses a mean-centered estimator equivalent up to a constant rescaling to an RLOO-style advantage, reducing bias toward low-variance prompts and length-induced distortions while preserving the GRPO update structure.

**Benchmarks and Setup.** We rigorously evaluate our method on two authoritative tool-use benchmarks. The Berkeley Function Calling Leaderboard (BFCL) (Patil et al., 2023) provides a standardized assessment of multi-step function calling and related capabilities, while API-Bank (Li et al., 2023) is a comprehensive three-tiered suite for evaluating tool invocation in multi-turn dialogues. To isolate the contribution of our algorithmic design, we fine-tune several open-source models and compare them against strong, domain-specialized baselines under matched training conditions.

**Performance Results.** As reported in Figure 2, Table 1 and Table 2, fine-tuning the Qwen3 model family with ResT establishes a new state-of-the-art on both benchmarks. The method yields significant performance gains over the GRPO baseline, with improvements of up to $8.76\%$ on BFCL and $3.02\%$ on API-Bank under an identical data budget. Notably, the Qwen3-4B-2507 model fine-tuned with ResT even surpasses GPT-4o by $0.38\%$ on multi-turn overall tool-use tasks, especially by $1.50\%$ on multi-turn base tasks. On the single-turn tool-use tasks, it leads by $4.11\%$. These behaviors indicate emergent forms of proactivity and metacognition, which enhance efficiency, mitigate hallucinations, and signal the emergence of foundational agentic intelligence.

Table 2: **API-Bank Test Results.** The results presented correspond to the highest score achieved by each method with its optimal hyperparameter settings. The evaluation dataset consists of 399 samples for Level 1, 67 for Level 2, and 131 for Level 3. Best in **bold**.

| Model | Overall Acc | Level 1 Acc | Level 2 Acc | Level 3 Acc |
|---|---|---|---|---|
| Qwen3-1.7B (Base) | 47.57% | 53.38% | 28.36% | 39.69% |
| Qwen3-1.7B (SFT) | 58.96% | 66.92% | 52.24% | 38.17% |
| Qwen3-1.7B (TSFT) | 54.77% | 61.4% | 43.28% | 40.46% |
| Qwen3-1.7B (RSFT) | 61.81% | 67.92% | 59.7% | 44.27% |
| Qwen3-1.7B (GRPO) | 63.65% | 70.68% | 61.19% | 41.22% |
| Qwen3-1.7B (SFT+GRPO) | 59.46% | 65.91% | 58.21% | 40.46% |
| Qwen3-1.7B (Dr.GRPO) | 57.55% | 84.06% | 73.66% | 15.25% |
| Qwen3-1.7B (Ours, ResT) | **64.99%** | 70.18% | 65.67% | 48.85% |
| Qwen3-4B-2507 (Base) | 66.33% | 72.68% | 64.18% | 48.09% |
| Qwen3-4B-2507 (SFT) | 60.30% | 67.92% | 59.70% | 37.40% |
| Qwen3-4B-2507 (TSFT) | 66.83% | 72.93% | 64.18% | 49.62% |
| Qwen3-4B-2507 (RSFT) | 64.49% | 69.92% | 59.7% | 50.38% |
| Qwen3-4B-2507 (GRPO) | 65.33% | 71.93% | 62.69% | 46.56% |
| Qwen3-4B-2507 (SFT+GRPO) | 68.51% | 74.44% | 65.67% | 51.91% |
| Qwen3-4B-2507 (Dr.GRPO) | 65.66% | 71.68% | 61.19% | 49.62% |
| Qwen3-4B-2507 (Ours, ResT) | **68.68%** | 73.68% | 64.18% | 55.73% |
| Qwen3-8B (Base) | 63.32% | 70.68% | 53.73% | 45.80% |
| Qwen3-8B (SFT) | 60.64% | 69.17% | 56.72% | 36.64% |
| Qwen3-8B (TSFT) | 59.80% | 66.42% | 50.75% | 44.27% |
| Qwen3-8B (RSFT) | 60.47% | 67.42% | 53.73% | 42.75% |
| Qwen3-8B (GRPO) | 66.50% | 75.44% | 64.18% | 40.46% |
| Qwen3-8B (SFT+GRPO) | 63.82% | 72.18% | 52.22% | 42.75% |
| Qwen3-8B (Dr.GRPO) | 68.15% | 88.00% | 79.25% | 38.12% |
| Qwen3-8B (Ours, ResT) | **70.69%** | 74.94% | 65.67% | 60.31% |
| Qwen3-14B (Base) | 65.49% | 71.68% | 56.72% | 51.15% |
| Qwen3-14B (SFT) | 67.34% | 76.44% | 62.69% | 41.98% |
| Qwen3-14B (TSFT) | 64.32% | 68.67% | 59.70% | 53.44% |
| Qwen3-14B (RSFT) | 64.99% | 70.68% | 55.22% | 52.67% |
| Qwen3-14B (GRPO) | 66.33% | 73.18% | 59.70% | 48.85% |
| Qwen3-14B (SFT+GRPO) | 67.34% | 73.43% | 58.21% | 53.44% |
| Qwen3-14B (Dr.GRPO) | 69.01% | 74.44% | 61.19% | 56.49% |
| Qwen3-14B (Ours, ResT) | **69.35%** | 73.43% | 62.69% | 60.31% |

Table 3: **API-Bank Test Results for Relaxations.** No dynamic reward denotes no dynamic scaling reward value. No gradients for CoT denotes no gradient update for chain-of-thought. No curriculum learning assigns a constant weight to each segment. Best in **bold**.

| Model | Overall Acc | Level 1 Acc | Level 2 Acc | Level 3 Acc |
|---|---|---|---|---|
| Qwen3-1.7B (*No dynamic reward*) | 64.15% | 69.92% | 58.21% | 49.62% |
| Qwen3-1.7B (*No gradients for CoT*) | 62.98% | 68.92% | 62.69% | 45.04% |
| Qwen3-1.7B (*No curriculum learning*) | 61.14% | 67.92% | 58.21% | 41.98% |
| Qwen3-1.7B (Ours, ResT) | **64.99%** | 70.18% | 65.67% | 48.85% |
| Qwen3-4B-2507 (*No dynamic reward*) | 64.15% | 71.18% | 64.18% | 42.75% |
| Qwen3-4B-2507 (*No gradients for CoT*) | 64.49% | 71.18% | 59.70% | 46.56% |
| Qwen3-4B-2507 (*No curriculum learning*) | 67.84% | 73.18% | 64.18% | 53.44% |
| Qwen3-4B-2507 (Ours, ResT) | **68.68%** | 73.68% | 64.18% | 55.73% |
| Qwen3-8B (*No dynamic reward*) | 64.15% | 71.68% | 64.18% | 41.22% |
| Qwen3-8B (*No gradients for CoT*) | 66.33% | 74.44% | 65.67% | 41.98% |
| Qwen3-8B (*No curriculum learning*) | 65.83% | 73.68% | 62.69% | 43.51% |
| Qwen3-8B (Ours, ResT) | **70.69%** | 74.94% | 65.67% | 60.31% |

## 4.3 ABLATION STUDY

Ablations across base models (Table 3) show that each ResT component is necessary for peak performance. Disabling dynamic rewards, CoT policy gradients, or the curriculum consistently reduces accuracy. On Qwen3-8B, removing dynamic rewards lowers accuracy by 6.54%, and dropping CoT

Table 4: **Ablation study on ResT initialization strategies and curriculum alignment** on the BFCL benchmark with Qwen3-1.7B. SFT warm-starting combined with a *Tuned Curriculum* (Tuned Curr.) achieves the best performance, demonstrating a synergistic effect. Average performance is calculated using the official scripts.

| Model | Overall Acc | Base | Miss Func | Miss Param | Long Context |
|---|---|---|---|---|---|
| Base | 8.62% | 12.00% | 5.50% | 9.50% | 7.50% |
| SFT | 14.25% | 17.50% | 14.50% | 14.50% | 10.50% |
| ResT(Cold Start) | 15.75% | 22.50% | 16.50% | 13.50% | 10.50% |
| SFT+ResT(Original Curr.) | 15.25% | 19.50% | 16.50% | 13.00% | 12.00% |
| SFT+ResT(Tuned Curr.) | **17.62%** | 23.50% | 19.00% | 14.00% | 14.00% |

policy gradients and the curriculum yields declines of 4.36% and 4.86%, respectively. These results indicate complementary contributions and explain ResT's robustness.

The ResT-trained model demonstrates robust generalization beyond its multi-turn training focus. As shown in Table D.1, our method performs comparably to the baseline on single-turn tasks and exhibits superior performance on live (real-time updated) tasks, highlighting its ability to generalize across different interaction regimes and execution environments. Considering that SFT cold-start alters the entropy distribution, we conduct a controlled comparison on the 4B model using both the original weights and a variant with upweighted reasoning and tool-parameter terms in Table 4, demonstrating that our entropy-aware weighting effectively bridges the post-SFT model. We also extend our evaluation to the Llama-3.2-3B-Instruct fine-tuned with ResT in Table D.2 and Table D.3 to verify that the effectiveness of our method is not confined to a specific model family. This confirms that the benefits of our approach are model-agnostic.

## 5 CONCLUSION

We study tool-use in LLM agents through the lens of RL and establish a theoretical link between token-level policy entropy and training stability: lowering entropy tightens variance bounds on policy gradients, and reward mass concentrates on structured, low-entropy tokens such as tool names, arguments, and output-format markers. Building on this insight, we propose ResT, an entropy-aware, token-level RL algorithm that reshapes policy gradients via entropy-informed reweighting and a lightweight curriculum that progressively upweights reasoning tokens. This enables a smooth shift from structural correctness to semantic reasoning and stabilizes convergence in long-horizon, multi-turn tool use. Empirically, ResT attains state-of-the-art results on BFCL and API-Bank, improving prior methods by up to $8.76\%$; when fine-tuned on a 4B base LLM, it further surpasses GPT-4o by $4.11\%$ on single-turn tasks and $1.50\%$ on multi-turn base tasks.

Our experiments concentrate on schema-constrained tool-use benchmarks. Within this setting, ResT implements a principled, entropy-aware reshaping of token-level policy gradients, translating our variance analysis into a practical curriculum over structural and reasoning tokens. A natural next step is to carry the same design philosophy to more open-ended language-agent tasks with learned or preference-based rewards, replacing rule-based execution scores with scalar feedback from reward models or LLM-as-a-judge evaluators applied to the final answer or the quality of the reasoning trace, while preserving the core mechanism of entropy-guided token reweighting. We expect these extensions to further broaden the applicability of ResT.

ACKNOWLEDGMENTS

This work is supported by National Natural Science Foundation of China (Grant No. 62550062 and 62576338), the Strategic Priority Research Program of Chinese Academy of Sciences (Grant No. XDA0480302), Beijing Natural Science Foundation (Grant No. L252145), and Young Scientists Fund of the State Key Laboratory of Multimodal Artificial Intelligence Systems (Grant No. ES2P100116).

ETHICS STATEMENT

We propose ResT to improve the training efficiency and stability of large language models (LLMs) acting as tool-use agents. We recognize that methods which increase agentic capability may entail dual-use risks. In particular, a more capable tool-using model, if misapplied, could facilitate malicious tasks, including automating harmful behaviors or attempting to access information without authorization. Our design choices and evaluation protocols therefore emphasize safety, bounded generalization, and auditability.

All experiments were conducted in a controlled academic setting on public benchmarks (BFCL and API-Bank) whose tools and APIs are simulated; no real-world services were invoked, and no sensitive or private data were accessed. We document data provenance, filtering, and preprocessing; exclude any potentially identifying content; and release scripts that regenerate the exact evaluation inputs. To reduce misuse risk, we enforce strict rate limits in evaluation, log all tool calls, and disable any actions that could affect external systems. Model selection and reporting follow a pre-specified protocol, and we provide seeds, prompts, and run logs to enable independent verification.

Our goal is to contribute a robust (Duan et al., 2026) and efficient training paradigm for reinforcement learning with LLM agents while advancing understanding of safe and controllable agent behavior. We encourage subsequent researchers and practitioners to assess societal and ethical impacts—including distributional harms, failure modes under domain shift, and cascading error risks—and to adopt responsible release practices. Upon acceptance, we intend to release model checkpoints and inference artifacts under a research-only license, accompanied by a detailed model card describing intended use, limitations, and safety considerations, as well as red-teaming guidance and an abuse-reporting channel.

REPRODUCIBILITY STATEMENT

To ensure full reproducibility, we release the complete implementation of ResT—including training and evaluation code, experiment scripts, and configuration files—on an anonymized GitHub repository:`https://github.com/1229095296/ResT_Tool_use_LLM`.

The repository specifies exact dependency versions, provides an environment requirement file, and pins all results to tagged commits. We include turnkey run scripts for every table and figure, along with hardware specifications, random seeds, and expected wall-clock times. Logs, intermediate artifacts, and checkpoints necessary to reproduce the reported numbers are stored with checksums; unit tests cover core components of the training loop and evaluation harness. Our experiments build on several open-source LLMs, and the repository documents model choices, preprocessing steps, and all implementation and training details.

We evaluate on two widely used public benchmarks, BFCL and API-Bank, both of which provide simulated tools/APIs rather than real services. For robustness, we report the best run for baselines and for ResT, using identical evaluation protocols. We provide full hyperparameter sweeps, ablation studies, and sensitivity analyses (e.g., to reward scaling, rollout length, and optimizer settings). All hyperparameters, data preprocessing pipelines, and experimental configurations appear in the appendix and in the repository documentation. Any deviations from benchmark defaults are explicitly noted, and we include scripts to validate dataset integrity and to regenerate all results from scratch.

To further support open science, upon acceptance we plan to release the trained model checkpoints and inference-time artifacts under a research license, accompanied by model cards that describe intended use, limitations, and safety considerations. We will also provide a minimal, dependency-

pinned inference package and reproducible notebooks to facilitate downstream evaluation and comparison.

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

# A SUPPLEMENTARY TRIAL

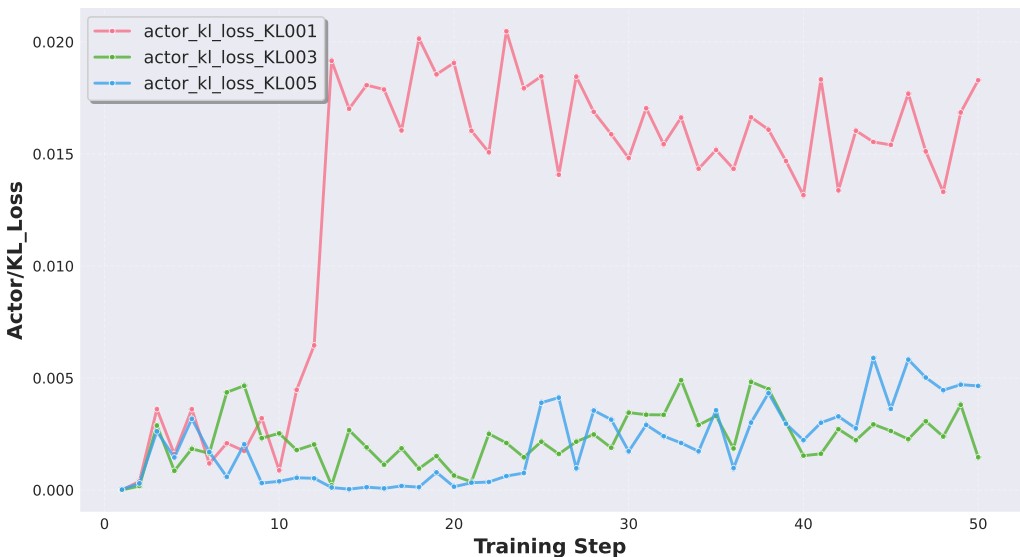

Figure A.1: Ablation Study on ResT: KL Loss Dynamics under Varying Coefficients.

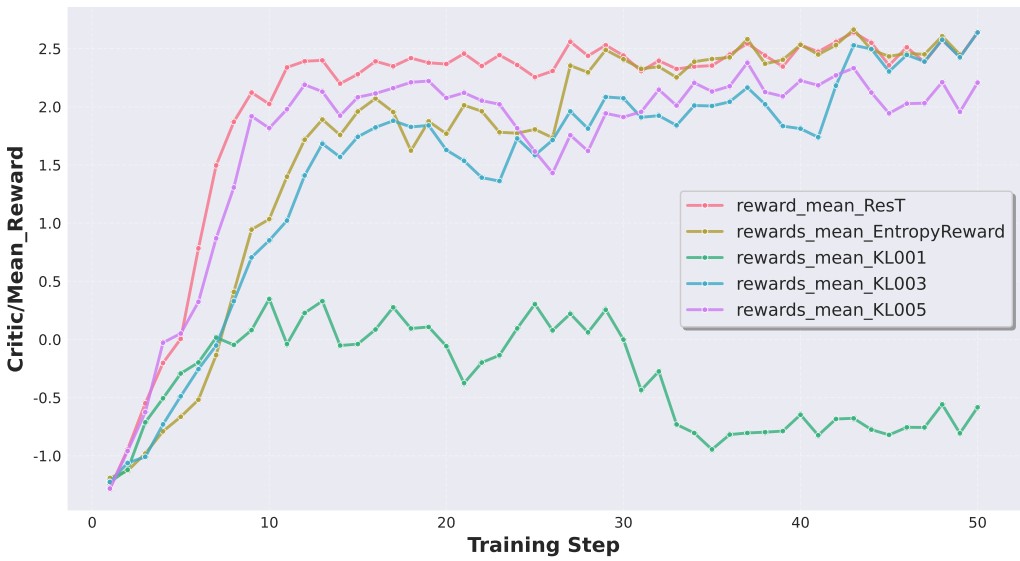

Figure A.2: Ablation Study on ResT: Impact of Regularization on Mean Training Reward.

Table A.1: Impact of Regularization Components on ResT Performance on the BFCL Multi-Turn Benchmark. Metrics computed with official scripts.

| Model | Overall Acc | Base | Miss Func | Miss Param | Long Context |
|---|---|---|---|---|---|
| Qwen3-4B-2507 (ResT with KL loss, $\beta = 0.001$) | 48.00% | 57.00 % | 52.50% | 34.00% | 48.50% |
| Qwen3-4B-2507 (ResT with KL loss, $\beta = 0.003$) | 50.25% | 59.50 % | 53.00% | 34.00% | 48.50% |
| Qwen3-4B-2507 (ResT with KL loss, $\beta = 0.005$) | 49.62% | 62.50 % | 53.00% | 37.00% | 51.50% |
| Qwen3-4B-2507 (ResT with entropy reward) | 47.88% | 58.50 % | 46.50% | 36.00% | 50.50% |
| Qwen3-4B-2507 ( ResT) | **50.38**% | 62.50% | 56.00% | 33.00% | 50.00% |

Table A.2: Statistical Validation with Multiple Seed Runs on Qwen3-4B-2507 and Qwen3-8B trained with ResT and GRPO.

| Model | Overall Acc | Base | Miss Func | Miss Param | Long Context |
|---|---|---|---|---|---|
| ResT-4B-seed1 | 50.38 % | 62.50 % | 56.00% | 33.00% | 50.00 % |
| ResT-4B-seed2 | 49.00 % | 58.50 % | 54.50% | 34.00% | 49.00 % |
| ResT-4B-seed3 | 50.50 % | 61.50 % | 56.50% | 34.50% | 49.50 % |
| GRPO-4B-seed1 | 40.88 % | 47.00 % | 43.50% | 31.50% | 41.50 % |
| GRPO-4B-seed2 | 41.12 % | 48.50 % | 43.50% | 31.50% | 41.00 % |
| GRPO-4B-seed3 | 41.62 % | 48.50 % | 45.50% | 27.00% | 45.50 % |
| ResT-8B-seed1 | 40.00 % | 52.50 % | 43.50% | 28.50% | 35.50 % |
| ResT-8B-seed2 | 40.13 % | 50.50 % | 45.00% | 32.00% | 33.00 % |
| ResT-8B-seed3 | 41.88 % | 52.00 % | 45.50% | 33.50% | 36.50 % |
| GRPO-8B-seed1 | 37.62 % | 45.00 % | 39.50% | 34.50% | 31.50 % |
| GRPO-8B-seed2 | 36.00 % | 42.00 % | 42.00% | 28.50% | 31.50 % |
| GRPO-8B-seed3 | 36.62 % | 47.00 % | 39.00% | 32.00% | 28.50 % |

Table A.3: BFCL test results with different methods on Qwen3-32B. Non-Live and Live represent the single-turn settings. Conversely, the multi-turn evaluation comprises four distinct subsets: Base, Miss Function, Miss Parameter, and Long Context.

| Model | Overall | Non live | live | Base | Miss Func | Miss Param | Long Context |
|---|---|---|---|---|---|---|---|
| Base | 69.25% | 88.90% | 77.83% | 54.50% | 41.00% | 30.50% | 46.50% |
| GRPO | 70.32% | 89.19% | 77.70% | 59.50% | 47.00% | 32.00% | 45.50% |
| ResT | **71.69%** | 89.58% | 78.85% | 60.50% | 56.50% | 33.50% | 44.00% |

Table A.4: Abalation study on no gradients for CoT. BFCL multi-turn test results on different models.

| Model | Overall | Base | Miss Func | Miss Param | Long Context |
|---|---|---|---|---|---|
| 1.7B (ResT) | 16.00% | 20.00% | 16.00% | 15.00% | 13.00% |
| 1.7B (No gradients for CoT) | 15.62% | 22.50% | 13.50% | 12.00% | 14.50% |
| 4B (ResT) | 50.38% | 62.50% | 56.00% | 33.00% | 50.00% |
| 4B (No gradients for CoT) | 48.75% | 62.00% | 48.00% | 32.50% | 52.50% |
| 8B (ResT) | 40.13% | 50.50% | 45.00% | 32.00% | 33.00% |
| 8B (No gradients for CoT) | 37.62% | 45.00% | 43.00% | 31.50% | 31.50% |

## B PROOFS

This appendix provides detailed proofs of the theoretical results presented in Section 3.1.

### B.1 PROOF OF LEMMA 1 (POLICY GRADIENT VARIANCE DECOMPOSITION)

*Proof.* **Setup.** Let a trajectory be $\tau = (x, y_{0:T-1})$ sampled from $\pi_\theta$, with return $R(\tau)$. For trajectory index $i$, write $\tau_i$, $r_i := R(\tau_i)$, and $y_{i,t}$. The per-time-step score is $\nabla_\theta \log \pi_\theta(y_{i,t} \mid x, y_{i,<t})$.

**Step 1. Policy gradient with a baseline.** By the policy gradient theorem,

$$\nabla_\theta J(\theta) = \mathbb{E}_{\tau \sim \pi_\theta}\left[\sum_{t=0}^{T-1} \nabla_\theta \log \pi_\theta(y_t \mid x, y_{<t})\, R(\tau)\right].$$

Let $b_t = b_t(x, y_{<t})$ be any baseline that does not depend on the sampled action $y_t$ given the state/history. Then, conditioning on $(x, y_{<t})$,

$$\mathbb{E}_{y_t \sim \pi_\theta}\big[\nabla_\theta \log \pi_\theta(y_t \mid x, y_{<t})\, b_t\big] = b_t \sum_{y_t} \pi_\theta(y_t \mid x, y_{<t})\, \nabla_\theta \log \pi_\theta(y_t \mid x, y_{<t})$$

$$= b_t\, \nabla_\theta \sum_{y_t} \pi_\theta(y_t \mid x, y_{<t}) \; = \; 0.$$

Hence one may replace $R(\tau)$ by the advantage $A(\tau) := R(\tau) - \sum_t b_t$ inside the expectation without changing its value. In what follows we set $b_t \equiv 0$ for simplicity (equivalently, redefine $\mu, \sigma$ as the population moments of $A(\tau)$ if a baseline is used).

**Step 2. Population standardization and unbiased direction.** Consider the population-standardized advantage

$$\hat{A}_i \; = \; \frac{r_i - \mu}{\sigma}, \qquad \mu := \mathbb{E}[r], \;\; \sigma^2 := \mathrm{Var}(r),$$

where expectations are under the current policy $\pi_\theta$. Define the trajectory-level gradient

$$g_i \; = \; \hat{A}_i \sum_{t=0}^{T-1} \nabla_\theta \log \pi_\theta(y_{i,t} \mid x, y_{i,<t}).$$

Taking expectations and using $\mathbb{E}[r_i - \mu] = 0$,

$$\mathbb{E}[g_i] = \frac{1}{\sigma} \mathbb{E}\left[(r_i - \mu) \sum_t \nabla_\theta \log \pi_\theta(y_{i,t} \mid x, y_{i,<t})\right] = \frac{1}{\sigma} \nabla_\theta J(\theta).$$

Thus $g_i$ is an unbiased *direction* estimator for $\nabla_\theta J(\theta)$, up to a scalar factor $1/\sigma$. If instead one uses the centered (unnormalized) advantage $\hat{A}_i = r_i - \mu$, then $\mathbb{E}[g_i] = \nabla_\theta J(\theta)$ exactly. In practice the factor $1/\sigma$ can be absorbed into the learning rate.

**Step 3. Mini-batch variance decomposition.** Let $g_1, \ldots, g_G$ be i.i.d. copies of $g_i$ with finite second moments, and define the mini-batch estimator

$$\bar{g} \; = \; \frac{1}{G} \sum_{i=1}^{G} g_i.$$

Its covariance is

$$\mathrm{Cov}(\bar{g}) = \mathbb{E}\big[(\bar{g} - \mathbb{E}[g_1])(\bar{g} - \mathbb{E}[g_1])^\top\big] = \frac{1}{G^2} \sum_{i=1}^{G} \sum_{j=1}^{G} \mathrm{Cov}(g_i, g_j).$$

Independence gives $\mathrm{Cov}(g_i, g_j) = 0$ for $i \neq j$, hence

$$\mathrm{Cov}(\bar{g}) = \frac{1}{G^2} \sum_{i=1}^{G} \mathrm{Cov}(g_i, g_i) = \frac{1}{G} \mathrm{Cov}(g_1).$$

Taking the $k$-th diagonal entry yields, for any coordinate $k$,

$$\mathrm{Var}\big(\bar{g}^{(k)}\big) = \frac{1}{G}\,\mathrm{Var}\big(g_1^{(k)}\big),$$

proving equation 2.

**Step 4. Consistency checks.** All symbols are trajectory-indexed ($i$) or time-indexed ($t$) but not conflated. The expectation and variance are with respect to the data-generating policy $\pi_\theta$. Population moments $(\mu, \sigma)$ refer to returns under $\pi_\theta$ at the current parameters. The i.i.d. assumption holds when trajectories are drawn independently from $\pi_\theta$. $\qquad \square$

*Remark (batch normalization of advantages).* If one instead uses the *batch-normalized* statistic

$$\hat{A}_i = \frac{r_i - \mu_\mathcal{G}}{\sigma_\mathcal{G}}, \qquad \mu_\mathcal{G} := \frac{1}{G}\sum_{j=1}^{G} r_j, \;\; \sigma_\mathcal{G}^2 := \frac{1}{G}\sum_{j=1}^{G}(r_j - \mu_\mathcal{G})^2,$$

then $\{g_i\}$ are no longer independent because each $g_i$ is a function of the full batch. A direct expansion gives the exact decomposition

$$\mathrm{Cov}(\bar{g}) = \frac{1}{G}\mathrm{Var}(g_1) + \frac{G-1}{G}\,\mathrm{Cov}(g_1, g_2),$$

where $\mathrm{Cov}(g_1, g_2)$ denotes the cross-covariance induced solely by the shared batch statistics. Under mild moment and smoothness conditions, a delta-method (influence function) expansion of $(\mu_\mathcal{G}, \sigma_\mathcal{G})$ around $(\mu, \sigma)$ yields $\mathrm{Cov}(g_1, g_2) = O(1/G)$, so

$$\mathrm{Cov}(\bar{g}) = \frac{1}{G}\mathrm{Var}(g_1) + O\!\left(\frac{1}{G}\right),$$

and the bias of $\mathbb{E}[\bar{g}]$ relative to $\nabla_\theta J(\theta)/\sigma$ is $O(1/G)$. Both corrections vanish as $G \to \infty$, but for small $G$ they can be non-negligible in practice.

## B.2 PROOF OF LEMMA 2 (SECOND-ORDER MOMENT–ENTROPY CONNECTION)

*Proof.* **Setup and notation.** Fix a single trajectory. Let $z_t \in \mathbb{R}^V$ denote the logits, $p_t = \text{softmax}(z_t) \in \Delta^{V-1}$ the token distribution, and $e_{y_t}$ the one-hot vector of $y_t \sim p_t$. Define the score vector $s_t := e_{y_t} - p_t \in \mathbb{R}^V$ and the Jacobian $J_t := \partial z_t / \partial \theta \in \mathbb{R}^{V \times d}$. Then

$$\nabla_\theta \log \pi_\theta(y_t \mid x, y_{<t}) \;=\; J_t^\top s_t, \quad g_i \;=\; \hat{A}_i \sum_{t=0}^{T-1} J_t^\top s_t.$$

All expectations below are taken over the joint randomness of the trajectory and the policy unless explicitly conditioned.

**Step 1. Second-order moment control.** By the definition of vector variance,

$$\text{Var}(g_i) \;=\; \mathbb{E}\Big[\|g_i\|^2\Big] - \|\mathbb{E}[g_i]\|^2 \;\leq\; \mathbb{E}\Big[\|g_i\|^2\Big] \;=\; \mathbb{E}\Big[\hat{A}_i^2 \Big\| \sum_{t=0}^{T-1} J_t^\top s_t \Big\|^2\Big]. \qquad (*)$$

Condition on the trajectory prefix up to time $t$. Since

$$\mathbb{E}[s_t \mid x, y_{<t}] \;=\; 0,$$

the sequence $\{J_t^\top s_t\}_{t=0}^{T-1}$ forms a vector-valued martingale difference, and thus is $L^2$-orthogonal. Therefore

$$\mathbb{E}\Big[\Big\| \sum_{t=0}^{T-1} J_t^\top s_t \Big\|^2\Big] = \sum_{t=0}^{T-1} \mathbb{E}\big[\|J_t^\top s_t\|^2\big].$$

Applying the inequality $\|J_t^\top s_t\| \leq \|J_t\|_F \|s_t\|$ gives

$$\mathbb{E}\Big[\hat{A}_i^2 \Big\| \sum_{t=0}^{T-1} J_t^\top s_t \Big\|^2\Big] \;\leq\; \mathbb{E}[\hat{A}_i^2] \sum_{t=0}^{T-1} \mathbb{E}[\|J_t\|_F^2 \|s_t\|^2] \;=\; \mathbb{E}[\hat{A}_i^2] \sum_{t=0}^{T-1} \beta_t \, \mathbb{E}[\|s_t\|^2],$$

where we set

$$\beta_t \;:=\; \mathbb{E}\big[\|J_t\|_F^2\big] \quad \text{or more tightly} \quad \beta_t \;:=\; \mathbb{E}\big[\|J_t\|_F^2 (1 - e^{-H_t})\big] \;\geq\; 0,$$

depending on whether we insert the entropy factor at this stage. Either choice preserves the quadratic-in-weights structure used later; we adopt the tighter definition when convenient.

**Step 2. Exact expression for the expected score norm.** Condition on $p_t$. Since $s_t = e_{y_t} - p_t$ with $y_t \sim p_t$,

$$\|s_t\|^2 \;=\; \|e_{y_t}\|^2 - 2\langle e_{y_t}, p_t \rangle + \|p_t\|^2 \;=\; 1 - 2p_{t,y_t} + \sum_{v=1}^{V} p_{t,v}^2.$$

Taking expectation over $y_t \sim p_t$ removes the index:

$$\mathbb{E}_{y_t \sim p_t}[\|s_t\|^2 \mid p_t] \;=\; 1 - \sum_{v=1}^{V} p_{t,v}^2 \;=\; 1 - \|p_t\|_2^2. \qquad (\dagger)$$

**Step 3. Entropy connection via the ordering $H_2 \leq H$.** Since the Rényi-2 entropy $H_2(p_t) = -\log \sum_v p_{t,v}^2$ is always upper-bounded by the Shannon entropy $H_t$, we obtain

$$H_t \;=\; \mathbb{E}_{v \sim p_t}\big[-\log p_{t,v}\big] \;\geq\; -\log \mathbb{E}_{v \sim p_t}[p_{t,v}] \;=\; -\log\Big(\sum_{v=1}^{V} p_{t,v}^2\Big).$$

Equivalently,

$$\sum_{v=1}^{V} p_{t,v}^2 \;\geq\; e^{-H_t}. \qquad (\ddagger)$$

Combining equation † and equation ‡ gives the entropy upper bound

$$\mathbb{E}[\|s_t\|^2 \mid p_t] \;=\; 1 - \|p_t\|_2^2 \;\leq\; 1 - e^{-H_t}.$$

Taking expectation over the randomness in $p_t$ yields

$$\mathbb{E}[\|s_t\|^2] \;\leq\; 1 - \mathbb{E}[e^{-H_t}] \;\leq\; 1 - e^{-\mathbb{E}[H_t]},$$

where the last inequality uses convexity of $\exp(\cdot)$ (optional if one keeps $1 - \mathbb{E}[e^{-H_t}]$). Equality in equation ‡ holds iff $p_t$ is uniform on its support, certifying tightness.

**Step 4. Conclusion.** Substituting the entropy bound for $\mathbb{E}[\|s_t\|^2]$ into equation $*$ gives

$$\mathrm{Var}(g_i) \;\leq\; \mathbb{E}[\hat{A}_i^2] \sum_{t=0}^{T-1} \mathbb{E}\big[\|J_t\|_F^2 \, \|s_t\|^2\big] \;\leq\; \mathbb{E}[\hat{A}_i^2] \sum_{t=0}^{T-1} \mathbb{E}\big[\|J_t\|_F^2 (1 - e^{-H_t})\big],$$

which is precisely the claimed second-order moment–entropy relation.

*Remarks on assumptions and tightness.* (i) The step factoring $\mathbb{E}[\hat{A}_i^2]$ is exact if $\hat{A}_i$ is independent of $(J_t, s_t)$ or after conditioning on the trajectory prefix; without independence one may apply Hölder to absorb the mismatch into a constant, leaving the functional dependence on entropy unchanged. (ii) The bound $\mathbb{E}[\|s_t\|^2] = 1 - \|p_t\|_2^2 \leq 1 - e^{-H_t}$ becomes tight when $p_t$ is uniform over its support, reflecting the ordering $H_2 \leq H$ between Rényi-2 and Shannon entropies. $\qquad\square$

### B.3 Proof of Theorem 2 (Optimal Entropy-Aware Reweighting)

*Proof.* **Step 1. Starting point.** By Lemma 2 and the inequality $\|J_t^\top s_t\|^2 \leq \|J_t\|_F^2 \|s_t\|^2$,

$$\mathrm{Var}(g_i) \leq \mathbb{E}\Big[\hat{A}_i^2 \sum_{t=0}^{T-1} \|J_t\|_F^2 \|s_t\|^2\Big].$$

Applying the entropy bound $\mathbb{E}[\|s_t\|^2] \leq 1 - e^{-H_t}$ gives

$$\mathrm{Var}(g_i) \leq \mathbb{E}[\hat{A}_i^2] \sum_{t=0}^{T-1} \mathbb{E}[\|J_t\|_F^2(1 - e^{-H_t})] = \mathbb{E}[\hat{A}_i^2] \sum_{t=0}^{T-1} \beta_t,$$

where we set

$$\beta_t := \mathbb{E}[\|J_t\|_F^2(1 - e^{-H_t})] \geq 0.$$

*Remark.* Factoring out $\mathbb{E}[\hat{A}_i^2]$ tacitly assumes that $\hat{A}_i$ is independent of $(J_t, H_t)$ or that the bound is taken conditionally on the trajectory; without independence one may invoke Hölder/Cauchy–Schwarz to obtain an equivalent quadratic objective in the weights, which preserves the optimal proportionality derived below.

**Step 2. Effect of reweighting.** Introduce per-step nonnegative weights $\tilde{w}_t \geq 0$ with normalization $\sum_{t=0}^{T-1} \tilde{w}_t = T$, and consider the reweighted estimator

$$g_i^{(\mathrm{rw})} = \hat{A}_i \sum_{t=0}^{T-1} \tilde{w}_t J_t^\top s_t.$$

Variance scales quadratically in the weights, hence

$$\mathrm{Var}(g_i^{(\mathrm{rw})}) \leq \mathbb{E}[\hat{A}_i^2] \sum_{t=0}^{T-1} \beta_t (\tilde{w}_t)^2.$$

**Step 3. Convex program.** We minimize the separable quadratic

$$\min_{\{\tilde{w}_t\}} \sum_{t=0}^{T-1} \beta_t (\tilde{w}_t)^2 \quad \text{s.t.} \quad \sum_{t=0}^{T-1} \tilde{w}_t = T, \quad \tilde{w}_t \geq 0.$$

If $\beta_t > 0$ for all $t$, the objective is strictly convex and the feasible set is a simplex, so the minimizer is unique.

**Step 4. Lagrangian and stationarity.** We first solve the equality-constrained problem (the inequality will be verified inactive). The Lagrangian reads

$$\mathcal{L}(\tilde{w}, \lambda) = \sum_{t=0}^{T-1} \beta_t (\tilde{w}_t)^2 + \lambda\Big(\sum_{t=0}^{T-1} \tilde{w}_t - T\Big).$$

Taking the derivative with respect to $\tilde{w}_t$ gives the stationarity condition

$$\frac{\partial \mathcal{L}}{\partial \tilde{w}_t} = 2\beta_t \tilde{w}_t + \lambda = 0 \quad \Longrightarrow \quad \tilde{w}_t = -\frac{\lambda}{2\beta_t} = \frac{c}{\beta_t} \quad \text{with } c := -\lambda/2.$$

**Step 5. Normalization and interiority.** Enforcing $\sum_{t=0}^{T-1} \tilde{w}_t = T$ yields

$$c \sum_{t=0}^{T-1} \beta_t^{-1} = T \quad \Longrightarrow \quad c = \frac{T}{\sum_{u=0}^{T-1} \beta_u^{-1}}, \qquad \lambda^* = -\frac{2T}{\sum_{u=0}^{T-1} \beta_u^{-1}} < 0.$$

Hence, for all $t$,

$$\tilde{w}_t^\star = \frac{T\beta_t^{-1}}{\sum_{u=0}^{T-1} \beta_u^{-1}} > 0,$$

which a posteriori verifies that the nonnegativity constraints are inactive at the optimum (the KKT multipliers for the inequalities vanish). Strict convexity then guarantees uniqueness.

**Step 6. Degenerate case.** If some $\beta_{t_0} = 0$, any feasible allocation supported on $\{t : \beta_t = 0\}$ attains objective value 0, which is minimal. Otherwise, when all $\beta_t > 0$, the interior solution above is the unique minimizer.

**Step 7. Optimal value and variance bound.** Substituting $\tilde{w}^\star$ into the quadratic gives

$$\sum_{t=0}^{T-1} \beta_t (\tilde{w}_t^\star)^2 = \sum_{t=0}^{T-1} \beta_t \left( \frac{T \beta_t^{-1}}{\sum_{u=0}^{T-1} \beta_u^{-1}} \right)^2 = \frac{T^2}{\sum_{u=0}^{T-1} \beta_u^{-1}}.$$

Therefore,

$$\mathrm{Var}\big(g_i^{(\mathrm{rw})}\big) \;\leq\; \mathbb{E}\big[\hat{A}_i^2\big] \cdot \frac{T^2}{\sum_{t=0}^{T-1} \beta_t^{-1}}.$$

**Step 8. Dual certificate via Cauchy–Schwarz.** For any feasible $\{\tilde{w}_t\}$,

$$\Big( \sum_{t=0}^{T-1} \tilde{w}_t \Big)^2 = \Big( \sum_{t=0}^{T-1} \sqrt{\beta_t}\, \tilde{w}_t \cdot \beta_t^{-1/2} \Big)^2 \leq \Big( \sum_{t=0}^{T-1} \beta_t (\tilde{w}_t)^2 \Big) \Big( \sum_{t=0}^{T-1} \beta_t^{-1} \Big),$$

which implies

$$\sum_{t=0}^{T-1} \beta_t (\tilde{w}_t)^2 \;\geq\; \frac{T^2}{\sum_{t=0}^{T-1} \beta_t^{-1}},$$

with equality if and only if $\tilde{w}_t \propto \beta_t^{-1}$, matching $\tilde{w}_t^\star$ and certifying optimality. $\qquad\square$

## C  TRAINING ALGORITHM: REST WITH RULE-BASED REWARDS

---

**Algorithm 1:** Rule-Based Reward Computation

---

**Input:** Prediction $(N_P, P_P, \mathbf{v}_P)$; ground truth $(N_G, P_G, \mathbf{v}_G)$ for each tool; format template $\mathcal{T}$;
        training progress $\bar{\nu} \in (0, 1)$; weights $(\beta_a, \beta_f)$.
**Output:** $\mathcal{R}_{i,\text{final}}$.

**Format score**:
$\mathcal{S}_{i,\text{format}} \leftarrow \mathbf{1}[\text{prediction matches } \mathcal{T} \text{ exactly}]$
**Name score**:
$r_{\text{name}} \leftarrow \frac{|N_G \cap N_P|}{|N_G \cup N_P|}$
**Parameter-name score**:
$r_{\text{para}} \leftarrow \sum_{G_j \in G} \frac{|P_G(G_j) \cap P_P(G_j)|}{|P_G(G_j) \cup P_P(G_j)|}$
**Parameter-value score**:
$r_{\text{value}} \leftarrow \sum_{G_j \in G} \sum_{v \in \mathbf{v}_G(G_j)} \mathbf{1}[P_G(G_j)[v] = P_P(G_j)[v]]$
**Normalization**:
$Z \leftarrow 1 + |G| + \sum_{G_j \in G} |\mathbf{v}_G(G_j)|$
$\mathcal{S}_{i,\text{acc}} \leftarrow \dfrac{r_{\text{name}} + r_{\text{para}} + r_{\text{value}}}{Z}$
**Dynamic scaling**:
$\mathcal{R}_{i,\text{final}} \leftarrow (1 - \bar{\nu})(\beta_a \, \mathcal{S}_{i,\text{acc}} + \beta_f \, \mathcal{S}_{i,\text{format}})$
**return** $\mathcal{R}_{i,\text{final}}$

---

---

**Algorithm 2:** ResT: Reshaping Token-level Policy Gradients

---

**Input:** Prompt $x$; policy $\pi_\theta$ with frozen reference $\pi_{\theta_{\text{old}}}$; group size $G$; PPO clip $\epsilon$; stability constant $\delta > 0$; curriculum proportion $\nu \in [0, 1]$; clipping bounds $(w_{\min}, w_{\max})$; step sizes $(\alpha_f, \alpha_p, \alpha_t)$; entropy-based initializer (Eq. 9).

**Output:** Updated parameters $\theta$.

**Rollout:**
**for** $i = 1$ **to** $G$ **do**
  Sample response $y_i \sim \pi_\theta(\cdot \,|\, x)$ containing plain-text and tool-call segments
  Compute reward $\mathcal{R}_{i,\text{final}}$ via Alg. 1

**Group-normalized advantage:**

$\mu \leftarrow \frac{1}{G} \sum_{i=1}^{G} \mathcal{R}_{i,\text{final}}, \quad \sigma \leftarrow \sqrt{\frac{1}{G} \sum_{i=1}^{G} (\mathcal{R}_{i,\text{final}} - \mu)^2} + \delta$

**for** $i = 1$ **to** $G$ **do**
  $\hat{A}_i \leftarrow \dfrac{\mathcal{R}_{i,\text{final}} - \mu}{\sigma}$

**Token partition and initialization:**
`// Initialize global token weights (shared across samples)`
Let $T$ be a representative sequence length or maximum length
Initialize $\tilde{w}_t$ for all $t = 1, \ldots, T$ using entropy-based rule (Eq. 9)

**Curriculum reweighting (global schedule):**
`// Apply curriculum updates once, shared by all sequences`
**for** $t = 1$ **to** $T$ **do**
  $\tilde{w}_t \leftarrow \text{Clip}(\tilde{w}_t, w_{\min}, w_{\max})$

**for** $t \in \mathcal{I}_{\text{fmt}}$ **do**
  $\tilde{w}_t \leftarrow \text{Clip}(\tilde{w}_t - \alpha_f \nu, \; w_{\min}, w_{\max})$

**for** $t \in \mathcal{I}_{\text{name}}$ **do**
  $\tilde{w}_t \leftarrow w_{\max}$

**for** $t \in \mathcal{I}_{\text{para}}$ **do**
  $\tilde{w}_t \leftarrow \text{Clip}(\tilde{w}_t + \alpha_p \nu, \; w_{\min}, w_{\max})$

**for** $t \in \mathcal{I}_{\text{thk}}$ **do**
  $\tilde{w}_t \leftarrow \text{Clip}(\tilde{w}_t + \alpha_t \nu, \; w_{\min}, w_{\max})$

`// Normalize globally`
$\bar{w} \leftarrow \frac{1}{T} \sum_{t=1}^{T} \tilde{w}_t$
**for** $t = 1$ **to** $T$ **do**
  $\omega_t \leftarrow \dfrac{\tilde{w}_t}{\bar{w} + \delta}$

**PPO-style token-level objective and update:**
**for** $i = 1$ **to** $G$ **do**
  **for** $t = 1$ **to** $T^i$ **do**
    $r_{i,t} \leftarrow \dfrac{\pi_\theta(y_{i,t} \,|\, x, y_{i,<t})}{\pi_{\theta_{\text{old}}}(y_{i,t} \,|\, x, y_{i,<t})}$
    $g_{i,t} \leftarrow \min\left( r_{i,t} \hat{A}_i, \; \text{clip}(r_{i,t}, 1 - \epsilon, 1 + \epsilon) \hat{A}_i \right)$

$\mathcal{L}_{\text{ResT}}(\theta) \leftarrow -\frac{1}{G} \sum_{i=1}^{G} \sum_{t=1}^{T} -\frac{\omega_t}{T} \cdot g_{i,t}$
Update $\theta \leftarrow \theta - \eta \, \nabla_\theta \mathcal{L}_{\text{ResT}}(\theta)$

---

**Helper:** $\text{Clip}(u, a, b) = \min(\max(u, a), b)$. Sets $\mathcal{I}_{fmt}, \mathcal{I}_{name}, \mathcal{I}_{para}, \mathcal{I}_{thk}$ are obtained via deterministic token tagging of the generated sequence. Eq. 9 provides the entropy-based initializer for $\tilde{w}_t$.

# D    COMPLETE EXPERIMENTS RESULTS

Table D.1: **BFCL Benchmark Results on Single Turn Scenarios.** All metrics are calculated using the official scripts. **Rel.** and **Irrel.** denote the relevance-detection and irrelevance-detection subsets of BFCL, respectively.

| Model | Overall Acc | Non-live Acc | Live Acc | Rel. | Irrel. |
|---|---|---|---|---|---|
| Qwen3-1.7B (Base) | 77.70% | 83.27% | 72.95% | 83.33% | 85.49% |
| Qwen3-1.7B (SFT) | 78.13% | 82.90% | 74.06% | 72.22% | 78.02% |
| Qwen3-1.7B (TSFT) | 78.76% | 83.67% | 74.59% | 77.78% | 78.04% |
| Qwen3-1.7B (RSFT) | 77.99% | 82.46% | 74.19% | 66.67% | 76.53% |
| Qwen3-1.7B (GRPO) | 77.57% | 83.00% | 72.95% | 72.22% | 73.42% |
| Qwen3-1.7B (SFT+GRPO) | 78.17% | 85.31% | 72.10% | 66.67% | 76.94% |
| Qwen3-1.7B (Ours, ResT) | **78.59%** | 84.65% | 73.43% | 66.67% | 76.30% |
| Qwen3-4B-2507 (Base) | 83.41% | 87.98% | 79.52% | 77.78% | 79.18% |
| Qwen3-4B-2507 (SFT) | 84.66% | 89.04% | 80.94% | 83.33% | 81.90% |
| Qwen3-4B-2507 (TSFT) | 83.70% | 87.31% | 80.63% | 83.33% | 81.39% |
| Qwen3-4B-2507 (RSFT) | 83.85% | 87.69% | 80.59% | 88.89% | 82.02% |
| Qwen3-4B-2507 (GRPO) | 83.82% | 89.40% | 79.08% | 83.33% | 81.52% |
| Qwen3-4B-2507 (SFT+GRPO) | 84.54% | 89.46% | 80.36% | 77.78% | 80.22% |
| Qwen3-4B-2507 (Ours, ResT) | **84.58%** | 88.65% | 81.12% | 77.78% | 79.67% |
| Qwen3-8B (Base) | 83.26% | 88.81% | 78.54% | 77.78% | 79.08% |
| Qwen3-8B (SFT) | 77.78% | 82.58% | 73.70% | 72.22% | 75.62% |
| Qwen3-8B (TSFT) | 78.30% | 81.88% | 75.26% | 83.33% | 76.72% |
| Qwen3-8B (RSFT) | 77.16% | 81.08% | 73.83% | 77.78% | 77.38% |
| Qwen3-8B (GRPO) | **84.37%** | 88.50% | 80.85% | 83.33% | 82.24% |
| Qwen3-8B (SFT+GRPO) | 80.39% | 85.90% | 75.70% | 77.78% | 75.96% |
| Qwen3-8B (Ours, ResT) | 84.11% | 90.08% | 79.03% | 94.44% | 77.62% |
| Qwen3-14B (Base) | 83.74% | 88.17% | 79.96% | 72.22% | 81.01% |
| Qwen3-14B (SFT) | 74.35% | 77.31% | 71.83% | 77.78% | 69.95% |
| Qwen3-14B (TSFT) | 75.62% | 79.90% | 71.97% | 83.33% | 72.48% |
| Qwen3-14B (RSFT) | 72.66% | 71.81% | 73.39% | 55.56% | 85.44% |
| Qwen3-14B (GRPO) | 84.04% | 89.10% | 79.74% | 77.78% | 79.31% |
| Qwen3-14B (SFT+GRPO) | 81.17% | 86.56% | 76.59% | 77.78% | 75.41% |
| Qwen3-14B (Ours, ResT) | **84.63%** | 89.10% | 80.81% | 77.78% | 80.56% |

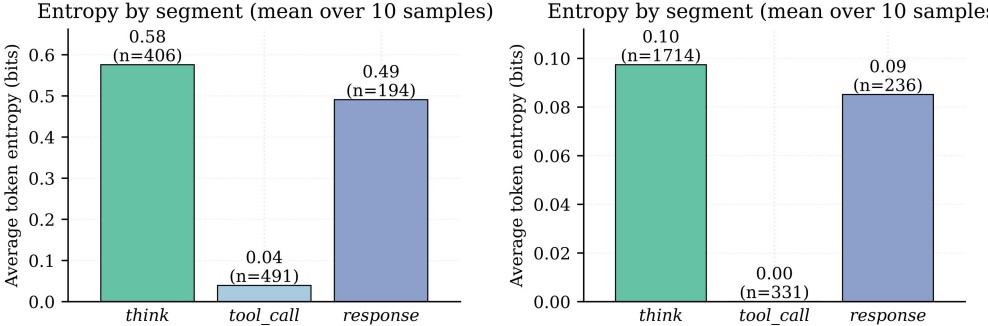

Figure D.1: Average region entropy on Llama-3.2-3B-Instruct (left) and Qwen3-1.7B (right). For each model, we randomly sample 10 instances and compute the entropy distribution over generated tokens. Tokens are partitioned into functional regions—reasoning, tool invocation, and final response.

Table D.2: **API-Bank Test Results on Llama3.2-3B-Instruct model.** We report the highest accuracy achieved for each method after hyper-parameter tuning, with a breakdown by difficulty level.

| Model | Overall Acc | Level 1 Acc | Level 2 Acc | Level 3 Acc |
|---|---|---|---|---|
| Llama-3.2-3B-Instruct(Base) | 37.35% | 43.61% | 32.84% | 20.61% |
| Llama-3.2-3B-Instruct(SFT) | 46.4% | 52.88% | 29.85% | 35.11% |
| Llama-3.2-3B-Instruct(TSFT) | 49.25% | 57.14% | 31.34% | 34.35% |
| Llama-3.2-3B-Instruct(RSFT) | 51.26% | 57.89% | 29.85% | 41.98% |
| Llama-3.2-3B-Instruct(GRPO) | 59.13% | 65.66% | 52.24% | 42.75% |
| Llama-3.2-3B-Instruct(SFT+GRPO) | 52.76% | 61.15% | 43.28% | 32.06% |
| Llama-3.2-3B-Instruct(Ours, ResT) | **62.81%** | 69.42% | 53.73% | 47.33% |

Table D.3: **BFCL Results on Llama3.2-3B-Instruct model.** All metrics are calculated using the official scripts. Overall accuracies for the three different tasks are provided separately. Best in **bold**.

| Model | Non-Live AST Acc | Live Acc | Multi Turn Acc | Relevance Detection | Irrelevance Detection |
|---|---|---|---|---|---|
| Llama-3.2-3B-Instruct (Base) | 17.44% | 43.85% | 0.00% | 77.78% | 66.07% |
| Llama-3.2-3B-Instruct (SFT) | 65.42% | 63.04% | 1.38% | 77.78% | 78.25% |
| Llama-3.2-3B-Instruct (TSFT) | 57.06% | 45.45% | 0.12% | 38.89% | 59.17% |
| Llama-3.2-3B-Instruct (RSFT) | 35.35% | 60.20% | 1.25% | 94.44% | 71.00% |
| Llama-3.2-3B-Instruct (GRPO) | 74.38% | 56.86% | 1.37% | 94.44% | 62.23% |
| Llama-3.2-3B-Instruct (SFT+GRPO) | 23.92% | 45.40% | 0.00% | 11.11% | 89.99% |
| Llama-3.2-3B-Instruct (Ours, ResT) | **83.19%** | **61.53%** | **3.50%** | 88.89% | 66.56% |

# E   GROUP RELATIVE POLICY OPTIMIZATION (GRPO)

Group Relative Policy Optimization (GRPO) serves as a critic-free, on-policy optimization algorithm designed for Large Language Models (LLMs). It presents an efficient alternative to Proximal Policy Optimization (PPO) by constructing a relative baseline for advantage estimation. This approach circumvents the need to learn a separate value network, thereby avoiding dependencies on methods like Generalized Advantage Estimation (GAE) which require a value function.

The core idea of GRPO is to dynamically construct a baseline for each sampled output using the reward distribution of a set of candidate outputs generated from the same input prompt. Specifically, for a given prompt $q$, we first sample a set of $S$ complete output sequences $\mathcal{O} = \{o_1, o_2, \ldots, o_S\}$ from the current policy $\pi_{\theta_{old}}$. A scalar reward $r_i = R(q, o_i)$ is then computed for each output $o_i$.

The empirical mean of these rewards serves as the baseline for evaluating each output, i.e., $b(q) = \text{mean}(\{r_1, \ldots, r_S\})$. The advantage for each rollout $o_i$ is then normalized using the standard deviation of the rewards to obtain the normalized advantage estimate $\hat{A}_i$:

$$\hat{A}_i = \frac{r_i - \text{mean}(\{r_1, \ldots, r_S\})}{\text{std}(\{r_1, \ldots, r_S\}) + \delta} \tag{20}$$

where $\delta$ is a small constant added for numerical stability to prevent division by zero. This "intra-group relative" advantage estimation effectively reduces reward variance.

During the policy update phase, this sequence-level advantage value $\hat{A}_i$ is applied to every timestep (token) within that sequence. GRPO performs policy updates by maximizing a clipped surrogate objective, analogous to PPO:

$$\mathcal{L}^{\text{GRPO}}(\theta) = \mathbb{E}_{q \sim \mathcal{D}, \, o \sim \pi_{\theta_{\text{old}}}} \left[ \sum_{t=1}^{|o|} \min\Big( \rho_t(\theta)\hat{A}, \ \text{clip}\big(\rho_t(\theta), \, 1-\epsilon, \, 1+\epsilon\big)\hat{A} \Big) \right.$$
$$\left. - \beta \, \text{KL}\big(\pi_\theta(\cdot|q) \big\| \pi_{\text{ref}}(\cdot|q)\big) \right]. \tag{21}$$

where $\rho_t(\theta) = \frac{\pi_\theta(o_t|q, o_{<t})}{\pi_{\theta_{old}}(o_t|q, o_{<t})}$ is the likelihood ratio of the new and old policies at timestep $t$. $\hat{A}$ corresponds to the normalized advantage of the current sequence $o$. The hyperparameters $\epsilon$ and $\beta$ control the clipping range and the strength of the KL-divergence penalty against a reference policy $\pi_{\text{ref}}$, respectively.

By adopting this formulation, GRPO retains the stability of PPO's trust-region updates while eliminating the complexities of training a value network, offering a simple yet effective framework for fine-tuning LLMs with reinforcement learning.

## F    TRAINING DETAILS

We conducted comprehensive experiments using the verl 0.5.0 framework, and the detailed training hyperparameters are provided in Table F.1 and F.2. The System prompt is shown in Figure G.1.

**Training Dynamics Analysis.** As illustrated in the training dynamics in Figure F.1, ResT exhibits a smoother and consistently lower policy entropy compared to GRPO. Regarding the mean response length, ResT shows a relatively small amplitude of change, whereas GRPO exhibits significant variation. Furthermore, ResT demonstrates stable convergence; the reward curve converges starting from step 30 and maintains this level through step 80, showing no signs of reward collapse.

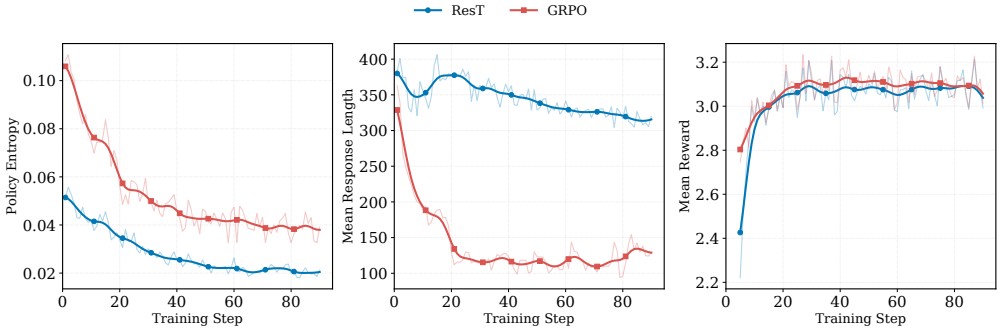

Figure F.1: Learning curves for ResT and GRPO during training steps. The training dynamics show that ResT achieves a significantly lower and smoother policy entropy compared to GRPO, while maintaining comparable reward performance and longer responses.

Table F.1: Configuration for ResT training on Qwen3 models.

| Data Configuration | |
| --- | --- |
| Train Batch Size | 512 |
| Validation Batch Size | 128 |
| Max Prompt Length | 2048 |
| Max Response Length | 2048 |
| **Optimization** | |
| Learning Rate | $1 \times 10^{-6}$ |
| PPO Mini Batch Size | 64 |
| KL Loss Used | False |
| **Rollout Configuration** | |
| Rollout Name | vllm 0.8.5 |
| GPUs | $8\times$ 141G H20 or $8\times$ 80G A100 |
| GPU Memory Utilization | 0.8 |
| Number of Rollouts | 4 or 8 |
| **Training & Logging** | |
| Save Frequency (Steps) | 15 |
| Test Frequency (Steps) | 15 |
| Total Epochs | 15 |
| **Initial Reweighting Factor** | |
| Initial Factor of Chain of Thoughts | 0.016 |
| Initial Factor of Tool Names | 1.2 |
| Initial Factor of Tool Parameters | 0.8 |
| Initial Factor of Output Format | 0.95 |
| **Curriculum Learning Configuration** | |
| $\alpha_f$ (update magnitude of output format) | 0.003 |
| $\alpha_p$ (update magnitude of output tool parameters) | 0.003 |
| $\alpha_t$ (update magnitude of reasoning) | $6 \times 10^{-5}$ |

Table F.2: Configuration for ResT training on Llama3.2-3B-Instruct.

| Data Configuration | |
|---|---|
| Train Batch Size | 512 |
| Validation Batch Size | 128 |
| Max Prompt Length | 2048 |
| Max Response Length | 2048 |
| **Optimization** | |
| Learning Rate | $1 \times 10^{-6}$ |
| PPO Mini Batch Size | 64 |
| KL Loss Used | False |
| **Rollout Configuration** | |
| Rollout Name | vllm 0.8.5 |
| GPUs | $4\times$ 80G A100 |
| GPU Memory Utilization | 0.8 |
| Number of Rollouts | 8 |
| **Training & Logging** | |
| Save Frequency (Steps) | 15 |
| Test Frequency (Steps) | 15 |
| Total Epochs | 15 |
| **Initial Reweighting Factor** | |
| Initial Factor of Chain of Thoughts | 0.6 |
| Initial Factor of Tool Names | 1.2 |
| Initial Factor of Tool Parameters | 0.6 |
| Initial Factor of Output Format | 1.0 |
| **Curriculum Learning Configuration** | |
| $\alpha_f$ (update magnitude of output format) | 0.002 |
| $\alpha_p$ (update magnitude of output tool parameters) | 0.002 |
| $\alpha_t$ (update magnitude of reasoning) | 0.002 |

# G  PROMPT DETAILS

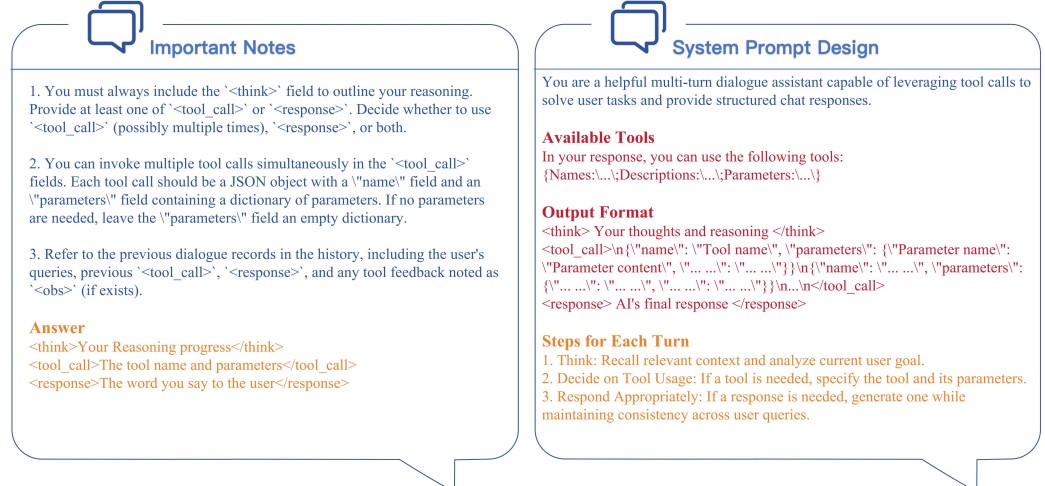

Figure G.1: The following Q&A template—comprising predefined instructions, guidelines, and constraints—standardizes model behavior and output formatting across training and evaluation. It ensures that, during training, the model learns consistently in line with target objectives, and that, during BFCL evaluation, performance is produced under a unified rule set.

# H    OUTPUT CASES

Table H.1: System Instruction Case 1 of ResT Models generalizing to irrelevant tool detection goals.

| User Prompt: |
| --- |
| **Dialogue Records History** |
| $< user >$ |
| Can you please modify my appointment scheduled for March 25th with Dr. Kim to March 26th with Dr. Lee? |
| $< /user >$ |
| $< response >$ |
| Sure, I can help you with that. Please provide me with the appointment ID and the new appointment date and doctor's name. |
| $< /response >$ |
| $< user >$ |
| The appointment ID is 34567890 and the new date is March 26th with Dr. Lee. |
| $< /user >$ |
| $< response >$ |
| Alright. I'll modify your appointment now. |
| $< /response >$ |
| $< user >$ |
| Based on our conversation above, please only make one tool call to solve my need. |
| $< /user >$ |
| |
| **Tools Available:** |
| In your response, you can use the following tools: |
| **QueryHealthData** |
| This API is used to query recorded health data for a given user within a specified time span. Parameters: |
| user_id (string): the identifier of the user, case-insensitive. |
| start_time (string): the start of the time span, formatted as YYYY-MM-DD HH:MM:SS. |
| end_time (string): the end of the time span, formatted as YYYY-MM-DD HH:MM:SS. |
| **CancelRegistration** |
| This API cancels the registration of a patient given an appointment ID. Parameters: |
| appointment_id (string): the identifier of the appointment. |
| **ModifyRegistration** |
| This API modifies the registration of a patient given an appointment ID. Parameters: |
| appointment_id (string): the identifier of the appointment. |
| new_appointment_date (string): the new appointment date, formatted as YYYY-MM-DD. |
| new_appointment_doctor (string): the new appointment doctor. |

Table H.2: Response case 1 of Qwen3-1.7B (ResT) generalizing to irrelevant tool detection goals.

$< think >$

Okay, let's see. The user wants to modify their appointment from March 25th with Dr. Kim to March 26th with Dr. Lee. The appointment ID provided is 34567890.

First, I need to check which tool to use. The available tools are QueryHealthData, CancelRegistration, and ModifyRegistration. The user is asking to modify an appointment, so ModifyRegistration is the right choice here.

Looking at the parameters for ModifyRegistration: appointment_id, new_appointment_date, and new_appointment_doctor. The user provided the appointment ID as 34567890, the new date as March 26th (which needs to be in %Y-%m-%d format, so 2023-03-26), and the new doctor as Dr. Lee.

I need to make sure the date format is correct. The user said March 26th, so the format should be 2023-03-26. The appointment ID is given as a string, so that's fine. The doctor's name is Dr. Lee, which is a string.

So the parameters would be: - appointment_id: "34567890" - new_appointment_date: "2023-03-26" - new_appointment_doctor: "Dr. Lee"

I should call ModifyRegistration with these parameters. No need for CancelRegistration since the user isn't canceling the old appointment. Just modifying it.

$< /think >$

$< tool\_call >$

"name": "ModifyRegistration", "parameters": "appointment_id": "34567890", "new_appointment_date": "2023-03-26", "new_appointment_doctor": "Dr. Lee"

$< /tool\_call >$

Table H.3: Response case 1 of Qwen3-4B-Thinking2507 (ResT) generalizing to irrelevant tool detection goals.

---

$< think >$

Okay, let's see. The user wants to modify their appointment from March 25th with Dr. Kim to March 26th with Dr. Lee. They provided the appointment ID as 34567890 and the new date is March 26th with Dr. Lee.

Wait, the previous response from the assistant said they'd modify the appointment now. But the user is now asking to only make one tool call. The available tools include ModifyRegistration. Let me check the parameters for that.

ModifyRegistration requires appointment_id, new_appointment_date (format %Y-%m-%d), and new_appointment_doctor. The user specified the new date as March 26th, which I need to convert to YYYY-MM-DD. March 26th would be 2024-03-26 assuming the current year, but maybe the system handles the date correctly. Wait, the user didn't specify the year, but the tool expects a specific format. Hmm, but the user said "March 26th" so the assistant should format it as 2024-03-26? Or maybe the API just takes the month and day without the year? Wait, the parameter description says the format is %Y-%m-%d, so it needs the year. But the user didn't mention the year. Wait, in the conversation history, the original appointment was on March 25th, so the new date is March 26th of the same year. Maybe the system assumes the current year, but since the user is talking about a specific date, the assistant should use the correct format.

Wait, the user's message says "the new date is March 26th with Dr. Lee." So the date is March 26th. The tool requires the date in YYYY-MM-DD. Let me think: in the context of the problem, the user is probably in the same year as the original appointment. Since the original was March 25th, maybe the current year is 2024. So March 26th would be 2024-03-26.

But the tool call needs to have the exact date string. Wait, the user hasn't specified the year, but in the problem context, maybe the system handles it. Alternatively, the assistant should just use the month and day as given. Hmm, but the tool requires the full date. Wait, the problem says the user is asking to make one tool call. The previous assistant said they'd modify the appointment now, so the next step is to call ModifyRegistration with the provided parameters.

The appointment_id is 34567890. The new_appointment_date is March 26th, which should be formatted as 2024-03-26 (assuming current year is 2024). The new_appointment_doctor is Dr. Lee.

Wait, the user said "the new date is March 26th with Dr. Lee." So the doctor's name is Dr. Lee. The tool's parameter for new_appointment_doctor is a string, so it's "Dr. Lee" or just "Lee"? The description says "the new appointment doctor", so probably the name as given. The user said "Dr. Lee", so the parameter should be "Dr. Lee".

So the tool call would be ModifyRegistration with appointment_id "34567890", new_appointment_date "2024-03-26", and new_appointment_doctor "Dr. Lee".

Wait, but the user's previous message says "the new date is March 26th with Dr. Lee." So the date is March 26th, but the year might be different. However, since the original appointment was March 25th (probably in the same year), the year is the same. Let's assume 2024 for the example.

So the parameters are:

appointment_id: "34567890"

new_appointment_date: "2024-03-26"

new_appointment_doctor: "Dr. Lee"

Therefore, the tool call should be to ModifyRegistration with those parameters.

$< /think >$

---

$< tool\_call >$

"name": "ModifyRegistration", "parameters": "appointment_id": "34567890", "new_appointment_date": "2024-03-26", "new_appointment_doctor": "Dr. Lee"

$< /tool\_call >$

Table H.4: Response case 1 of Qwen3-8B (ResT) generalizing to irrelevant tool detection goals.

$< think >$
Okay, the user wants to modify their appointment. Let me check the previous messages. They provided the appointment ID as 34567890, the new date is March 26th, and the new doctor is Dr. Lee. The tool required is ModifyRegistration. The parameters needed are appointment_id, new_appointment_date, and new_appointment_doctor. The date format should be YYYY-MM-DD. So March 26th becomes 2023-03-26. The doctor's name is Dr. Lee. I'll structure the tool call with these details.
$< /think >$

$< tool\_call >$
"name": "ModifyRegistration", "parameters": "appointment_id": "34567890", "new_appointment_date": "2023-03-26", "new_appointment_doctor": "Dr. Lee"
$< /tool\_call >$

Table H.5: Response case 1 of Llama3.2-3B-Instruct (ResT) generalizing to irrelevant tool detection goals.

$< think >$
Based on the previous conversation, we have the appointment ID and the new appointment date and doctor's name. We can use the ModifyRegistration tool to update the appointment details.
$< /think >$

$< tool\_call >$
"name": "ModifyRegistration", "parameters": "appointment_id": "34567890", "new_appointment_date": "2023-03-26", "new_appointment_doctor": "Dr. Lee"
$< /tool\_call >$

Table H.6: Response case 1 of Qwen3-14B (ResT) generalizing to irrelevant tool detection goals.

$< think >$

Okay, let's see. The user wants to modify their appointment. They mentioned the appointment ID is 34567890, the new date is March 26th, and the new doctor is Dr. Lee.

Looking at the tools available, the ModifyRegistration tool is the right one. It requires the appointment_id, new_appointment_date, and new_appointment_doctor.

The date needs to be in %Y-%m-%d format. March 26th would be 2023-03-26, but since the year isn't specified, maybe it's safe to assume the current year unless stated otherwise. However, the user didn't mention the year, so perhaps the system expects just the date without the year. Wait, the parameter specifies the format as %Y-%m-%d, so the year is required. Hmm, but the user only said "March 26th". Maybe the system allows omitting the year, but according to the tool's parameters, it's required. Wait, the user's original request was to change from March 25th to March 26th. The original appointment date wasn't specified with a year, but the tool requires a full date. This might be an issue. However, since the user hasn't provided the year, maybe the system expects it to be in the same year as the original appointment. But since the original appointment date wasn't given with a year, perhaps the assistant should use the same year as the current date. Alternatively, maybe the system is designed to handle dates without years, but according to the tool's parameters, the format is strict. Wait, the tool's parameter for new_appointment_date says the format is %Y-%m-%d. So the user must provide the year. But the user only said "March 26th". This could be a problem. However, in the previous response, the assistant said they would modify the appointment, so maybe they assumed the year is correct. Maybe the user is in the same year as the original appointment. Since the original appointment was on March 25th (without a year), perhaps the system is designed to use the current year. But since the user hasn't specified, maybe the assistant should proceed with the date as given, even if the year is missing. Wait, but the tool requires the full date. This might be an error. However, given the user's input, the assistant has to work with what's provided. Maybe the system is lenient, or the user expects the assistant to proceed with the given date. So, the assistant will format the date as 2023-03-26, assuming the current year. Alternatively, maybe the system allows just the month and day, but according to the tool's parameters, it's required to have the year. This is a bit conflicting, but since the user hasn't provided the year, perhaps the assistant should proceed with the information given, even if it's incomplete. But that might cause an error. However, the user's previous message said "March 26th with Dr. Lee," so the assistant has to use that. Maybe the system is designed to accept the date without the year, but the tool's parameter says it's required. Hmm, this is a problem. But given the context, the assistant has to proceed with the available information. So, the parameters would be appointment_id: 34567890, new_appointment_date: 2023-03-26 (assuming current year), new_appointment_doctor: Dr. Lee. The tool call would be ModifyRegistration with those parameters.

$< /think >$

$< tool\_call >$

"name": "ModifyRegistration", "parameters": "appointment_id": "34567890", "new_appointment_date": "2023-03-26", "new_appointment_doctor": "Dr. Lee"

$< /tool\_call >$

Table H.7: System Instruction Case 2 of ResT Models generalizing to irrelevant tool detection goals.

| User Prompt |
| --- |
| **Dialogue Records History** |

$< user >$
Could you get the current levels of nitrous oxide in the atmosphere for me?
$< /user >$
$< think >$
I should use the appropriate tool with proper parameters to respond to the user's need.
$< /think >$
$< tool\_call >$
"name": "Nitrous Ozone Levels API", "parameters": "nitrous": "current"
$< /tool\_call >$
$< obs >$
["name": "Nitrous Ozone Levels API", "results": "nitrousOxideLevels": "currentLevel": "330 ppb", "dateMeasured": "2020-10-09T06:00:00.000Z", "location": "Global", "dataSources": ["NASA Ozone Monitoring Instrument", "NOAA Global Monitoring Laboratory"], "error": null]
$< /obs >$
$< think >$
I should directly respond to the user's need.
$< /think >$
$< response >$
The current nitrous oxide levels in the atmosphere are as follows:

- **Current Level**: 330 parts per billion (ppb)

- **Date Measured**: 2020-10-09T06:00:00.000Z

- **Location**: Global

**Data Sources**

- NASA Ozone Monitoring Instrument

- NOAA Global Monitoring Laboratory

$< /response >$

| Tools Available |
| --- |

In your response, you can use the following tools:
**Nitrous Ozone Levels API**
Description: Retrieves nitrous oxide levels in the atmosphere, providing current and historical data. Parameters:
"nitrous\": {"description": "Optional filter for nitrous oxide levels (e.g., 'current', 'historical', or a specific value)", "type": "string", "default": ""}
**Country Information API**
Description: Retrieve information about countries, including their capitals, currencies, and flags. Parameters: {}
**M4.5+ Earthquakes, Past 7 Days**
Description: Retrieve a list of earthquakes with a magnitude of 4.5 or greater that have occurred in the past 7 days. Parameters:
"start_string\": {"description": "The start string of the 7-day period (in ISO format, e.g., 2023-03-01T00:00:00.000Z)", "type": "string", "default": ""}, "end_string\": {"description": "The end string of the 7-day period (in ISO format, e.g., 2023-03-08T00:00:00.000Z)", "type": "string", "default": ""}

Table H.8: System Instruction Case 2 of ResT Models generalizing to irrelevant tool detection goals (Continued).

| User Prompt |
| --- |
| $< user >$
Thank you for the information. It's alarming to see such high levels of nitrous oxide. Could you explain the impact of nitrous oxide on the environment and human health in more detail?
$< /user >$
$< think >$
I should directly respond to the user's need.
$< /think >$
$< response >$
Certainly. Nitrous oxide, often known as laughing gas, has several significant impacts on both the environment and human health.

**Environmental Impact:**

1. **Greenhouse Gas**: Nitrous oxide is over 300 times more potent as a greenhouse gas than carbon dioxide. It can trap significant amounts of heat in the atmosphere, thereby contributing to global warming and climate change.

2. **Ozone Layer Depletion**: Nitrous oxide is also the most significant ozone-depleting substance being emitted today. It breaks down and destroys the ozone layer, increasing the Earth's exposure to harmful ultraviolet (UV) radiation.

**Impact on Human Health:**

1. **Respiratory Issues**: Inhalation of nitrous oxide can lead to oxygen deprivation, which can affect both the brain and the heart. Reduced oxygen can lead to an array of respiratory issues, including shortness of breath, wheezing, and potential long-term harm to overall lung health.

2. **Neurological Issues**: Nitrous oxide is known to affect the nervous system. It can cause harmful neurological impacts, including dizziness, loss of motor control, and in severe cases, a risk of seizures or coma.

3. **Vitamin B12 Deficiency**: Prolonged exposure to or inhalation of nitrous oxide can inhibit the body's ability to use vitamin B12, leading to vitamin B12 deficiency. This can result in anemia and potential neurological damage.

Always be aware of your environment and attempt to reduce nitrous oxide emissions where possible. This can include practices like reducing motor vehicle emissions and being mindful of agricultural practices.
$< /response >$

$< user >$
Thank you for the detailed explanation. It's quite concerning how much nitrous oxide affects both our planet and our health. Could you also provide some information on which countries are the largest emitters of nitrous oxide?
$< /user >$ |

Table H.9: Response case 2 of Qwen3-1.7B (ResT) generalizing to irrelevant tool detection goals.

| |
|---|
| $< think >$
Okay, the user asked for information on the largest emitters of nitrous oxide. Let me think about the tools available. The Country Information API can retrieve country details, including capitals, currencies, and flags. But the user wants specific data on emissions, which the API might not provide directly. The API's description says it retrieves country information, but the parameters are empty. Hmm, maybe the API doesn't have emission data. The user might need data that's not covered by the available tools. Since the tools don't include a specific emissions API, I can't fetch that information. I should inform the user that the current tools can't provide that data and suggest they check official sources like the EPA or WHO. But wait, the user might expect the Country Information API to have some data. Let me check the API's description again. It says "Retrieve information about countries, including their capitals, currencies, and flags." There's no mention of emissions. So, the answer is that the tools can't provide emission data. I need to respond that accordingly.
$< /think >$ |
| $< response >$
The available tools do not provide specific data on nitrous oxide emissions by country. For detailed information on the largest emitters, you would need to consult official sources such as the Environmental Protection Agency (EPA) or the World Health Organization (WHO), which track greenhouse gas emissions globally. These organizations publish comprehensive reports on emissions by country, sector, and year.
$< /response >$ |

Table H.10: Response case 2 of Qwen3-4B-Thinking2507 (ResT) generalizing to irrelevant tool detection goals.

$< think >$ Okay, the user is asking about the largest emitters of nitrous oxide. Let me think.

First, I need to check which tool can provide country information. The available tools are Nitrous Ozone Levels API, Country Information API, and Earthquakes API. The Country Information API retrieves info like capitals, currencies, flags, but does it include emissions data?

Wait, the Country Information API's description says it's for country info including capitals, currencies, flags. It doesn't mention emissions. The Nitrous Ozone Levels API gives current levels but not country-specific data. The earthquakes tool is unrelated.

Hmm, the user is asking for the largest emitters of nitrous oxide. The tools provided don't have a specific one for emissions by country. The Country Information API might not have that data either.

Wait, maybe the user expects me to use the Country Information API but that's not the right tool here. The problem is that the available tools don't include a nitrous oxide emissions by country API. The user's question is beyond the scope of the provided tools.

In the previous interactions, the assistant used the Nitrous Ozone Levels API to get current levels, but that's global data. The user now wants country-specific emissions data, which isn't covered by the existing tools.

So, since none of the tools can retrieve country emissions data, I should inform the user that I can't provide that information with the current tools. I need to respond that I don't have access to that data and maybe suggest they check other sources.
$< /think >$

$< response >$
I don't have access to real-time or country-specific nitrous oxide emissions data through the available tools. The tools I can use (Nitrous Ozone Levels API, Country Information API, and Earthquake API) don't provide emissions statistics by country. For this type of environmental data, you might want to consult sources like the Environmental Protection Agency (EPA), the Intergovernmental Panel on Climate Change (IPCC), or the United Nations Framework Convention on Climate Change (UNFCCC). Would you like me to help with something else that I *can* address with the available tools?
$< /response >$

Table H.11: Response case 2 of Qwen3-8B (ResT) generalizing to irrelevant tool detection goals.

$< think >$ Okay, the user is asking about the largest emitters of nitrous oxide. Let me think. I remember that nitrous oxide emissions are influenced by factors like agriculture, industrial activities, and energy production. But wait, the tools available here are the Nitrous Ozone Levels API, Country Information API, and the Earthquakes API. Hmm, none of these tools directly provide emission data by country.

The Nitrous Ozone Levels API gives current levels but not emission sources. The Country Information API provides general country data but not specific emissions. The Earthquakes API is unrelated. So, I can't use the provided tools to get the exact countries with the highest emissions. I need to inform the user that I can't retrieve that specific information with the current tools. But maybe I can offer general knowledge instead. Wait, the user might be expecting an answer, so I should clarify the limitation and provide available alternatives or sources where they can find that info. Let me check if there's any way to use the existing tools indirectly. No, I don't think so. So the best approach is to apologize for the limitation and suggest reliable sources like the IPCC reports or environmental agencies.
$< /think >$

$< response >$
Unfortunately, I don't have access to real-time emission data or country-specific nitrous oxide emission statistics through the current toolset. However, I can share that agricultural activities (e.g., fertilizer use), industrial processes, and combustion of fossil fuels are major global sources of nitrous oxide emissions. For detailed country-level data, you might refer to reports from organizations like the Intergovernmental Panel on Climate Change (IPCC) or the United Nations Environment Programme (UNEP), which track greenhouse gas emissions globally.
$< /response >$

Table H.12: Response case 2 of Qwen3-14B (ResT) generalizing to irrelevant tool detection goals.

$< think >$
Okay, the user is asking which countries are the largest emitters of nitrous oxide. Let me think about the tools I have. The available tools are the Nitrous Ozone Levels API, Country Information API, and the M4.5+ Earthquakes API. None of these tools directly provide data on countries' emissions of nitrous oxide. The Nitrous Ozone Levels API gives current and historical levels but not by country. The Country Information API provides general country data but not emissions. The Earthquake API is unrelated.
Since none of the tools can answer the user's question, I need to inform them that I can't retrieve that specific information. I should apologize and explain that my current capabilities don't include accessing emission data by country. Maybe suggest they check sources like the IPCC or EPA for detailed reports. I need to make sure the response is helpful and guides them to reliable external resources.
$< /think >$

$< response >$
I appreciate your concern, and you're absolutely right that nitrous oxide emissions are a critical issue. However, I currently don't have access to tools or data sources that can directly retrieve country-specific nitrous oxide emission rankings. For detailed information on national emissions, I recommend consulting authoritative sources like: - **IPCC Reports** (Intergovernmental Panel on Climate Change) - **EPA's Greenhouse Gas Inventory** (U.S. Environmental Protection Agency) - **UNFCCC National Reports** (United Nations Framework Convention on Climate Change)
These organizations track emissions by country and provide breakdowns of sources like agriculture, industry, and fossil fuels. Would you like help finding links to these resources?
$< /response >$

Table H.13: Response case 2 of Llama3.2-3B-Instruct (ResT) generalizing to irrelevant tool detection goals.

$< think >$
To provide information on the largest emitters of nitrous oxide, I will use the Country Information API to gather data on countries that emit significant amounts of nitrous oxide.
$< /think >$

$< response >$
Unfortunately, the Country Information API does not provide specific information on nitrous oxide emissions. It primarily focuses on providing general information about countries, including their capitals, currencies, and flags. If you're looking for information on greenhouse gas emissions, I recommend consulting the United Nations Framework Convention on Climate Change (UNFCCC) or other reputable sources that track and report on greenhouse gas emissions by country.
$< /response >$

# I  USE OF LARGE LANGUAGE MODELS

We used a general-purpose large language model solely to polish the English writing of this manuscript—specifically grammar, wording, and stylistic clarity—and only in the Introduction and Conclusion sections, after all research design, experiments, analyses, and writing were completed. The model was not used to generate ideas, datasets, code, algorithms, proofs, or empirical results; it was not used for code synthesis or editing, image creation or editing, figure design, LaTeX equation or formula derivation, mathematical reasoning steps, data analysis, statistical testing, literature search, citation generation, or reference formatting. Interactions with the model were limited to surface-level copy-editing of author-written text. No proprietary, sensitive, or personal data were provided beyond our own draft sentences for editing. All suggested edits were reviewed, verified, and, when appropriate, further revised by the authors, who take full responsibility for the accuracy, originality, and integrity of the final manuscript.

