# OpenReview forum: "ResT: Reshaping Token-Level Policy Gradients for Tool-Use Large Language Models"
_ICLR.cc/2026/Conference — ICLR 2026 Poster_

### Official Review · Reviewer_Nagq · 2025-10-31

**Soundness:** 3
**Presentation:** 3
**Contribution:** 3
**Rating:** 6
**Confidence:** 3

**Summary:**

The work is a modification on GRPO that reshapes token-level policy gradients using entropy-aware weights computed over regions of the response that are of interest to a tool call, such as format tags, tool names, parameters, chain-of-thought, so that so lower-entropy, reward-critical tokens get higher weight. This is combined with a curriculum that increases weights on reasoning/parameter tokens as training progresses, smoothly shifting focus from structural correctness to sophistication of reasoning. Building on top of veRL, this technique achieves achieves state-of-the-art on BFCL and API-Bank.

**Strengths:**

* The work is reasonably well written with no major grammatical or formatting errors. The schematics are easy to understand and the paper is easy to read.
* The work theoretically establishes a principled connection between token-level entropy and gradient variance which grounds the approach well. I've checked the proofs roughly and they appear correct
* The problem is important (tool-call agents are of great interest in the community, and sparse reward is an on-going problem) and the essential ideas intuitively make sense (we need to get the calling conventions right before focusing on argument semantics)

**Weaknesses:**

The experimental results, though extensive, does not unequivocally establish ResT as a state of the art. It's unclear whether the accuracy gap falls within the variance or not (https://www.anthropic.com/research/statistical-approach-to-model-evals), and I cannot find an authoritative benchmark on either BFCL multi-turn or API-Bank that can support the SOTA claim. The results do look strong, just not conclusively strong.

**Questions:**

veRL is a complex codebase. I would appreciate the authors highlighting the changes they have made branching off of v0.5 to facilitate future replication and adoption efforts.

---

> ### Author Response · Authors · 2025-11-21
> **Response to Reviewer Nagq**
>
> We sincerely thank the reviewer for the encouraging assessment and the reference to statistical evaluation standards, which are crucial for distinguishing genuine algorithmic gains from stochastic variance. The main concerns are:
>
> * Clarification on whether the observed improvements exceed the range of random fluctuation (Q1).
> * Clarification on the SOTA designation and the standardization of the evaluation suite (Q2).
> * Requests for detailing modifications to veRL v0.5 to facilitate reproducibility (Q3).
>
> In response, we have conducted additional multi-seed experiments to quantify variance, clarified our SOTA claims and the standardization of the evaluation suite, and documented the modifications to veRL v0.5 to facilitate reproducibility.
>
>
> **_Q1: It's unclear whether the accuracy gap falls within the variance or not._**
>
> **_A1:_**
> We appreciate the reviewer’s focus on statistical significance and robustness. To verify that our reported gains exceed evaluation noise, we extend our experiments to three independent runs on BFCL for both Qwen3-4B and Qwen3-8B. We report mean $\pm$ std over seeds in **Table A2**:
>
> | Model | Overall Acc | Base | Miss Func | Miss Param | Long Context |
> |--------------------|-----------:|:------:|:---------:|:----------:|:------------:|
> | ResT-4B-seed1 | 50.38% | 62.50% | 56.00% | 33.00% | 50.00% |
> | ResT-4B-seed2 | 49.00% | 58.50% | 54.50% | 34.00% | 49.00% |
> | ResT-4B-seed3 | 50.50% | 61.50% | 56.50% | 34.50% | 49.50% |
> | GRPO-4B-seed1 | 40.88% | 47.00% | 43.50% | 31.50% | 41.50% |
> | GRPO-4B-seed2 | 41.12% | 48.50% | 43.50% | 31.50% | 41.00% |
> | GRPO-4B-seed3 | 41.62% | 48.50% | 45.50% | 27.00% | 45.50% |
> | ResT-8B-seed1 | 40.00% | 52.50% | 43.50% | 28.50% | 35.50% |
> | ResT-8B-seed2 | 40.13% | 50.50% | 45.00% | 32.00% | 33.00% |
> | ResT-8B-seed3 | 41.88% | 52.00% | 45.50% | 33.50% | 36.50% |
> | GRPO-8B-seed1 | 37.62% | 45.00% | 39.50% | 34.50% | 31.50% |
> | GRPO-8B-seed2 | 36.00% | 42.00% | 42.00% | 28.50% | 31.50% |
> | GRPO-8B-seed3 | 36.62% | 47.00% | 39.00% | 32.00% | 28.50% |
>
> On **Qwen3-4B**, we run three independent training and evaluation runs with different random seeds under the same BFCL multi-turn test set and decoding configuration. GRPO-4B achieves an average BFCL multi-turn overall accuracy of $41.21 \pm 0.38$, while ResT-4B reaches $49.96 \pm 0.83$, an absolute gain of **+8.75** points. A Welch $t$-test on these multi-turn scores yields $t = 16.57$ with approximately $df = 2.79$ and $p = 7.2 \times 10^{-4}$. The 95% confidence interval (CI) for the mean difference is $[7.00, 10.51]$; the 95% CIs for the individual means are approximately $[40.27, 42.14]$ for GRPO and $[47.89, 52.03]$ for ResT. Thus, on our primary metric of multi-turn tool use, the improvement of ResT over GRPO is far larger than the run-to-run variance and is highly unlikely to be explained by random fluctuation.
>
> For **Qwen3-8B**, we similarly perform three independent BFCL runs. GRPO-8B obtains $36.75 \pm 0.82$, while ResT-8B attains $40.67 \pm 1.05$, giving an absolute gain of **+3.92** points. Welch’s test yields $t = 5.11$, $df = 3.77$, and $p = 8.1 \times 10^{-3}$, with a 95% CI of $[1.74, 6.11]$ for the difference; the 95% CIs for the means are $[34.72, 38.78]$ for GRPO and $[38.06, 43.28]$ for ResT. These results indicate a statistically significant improvement for the 8B model as well.

---

> ### Author Response · Authors · 2025-11-21
> **Response to Reviewer Nagq**
>
> **_Q2: The experimental results, though extensive, do not unequivocally establish ResT as a state of the art. I cannot find an authoritative benchmark on either BFCL multi-turn or API-Bank that can support the SOTA claim._**
>
> **_A2:_**
> We appreciate the opportunity to clarify our evaluation protocol and the scope of our SOTA claim.
> We use **BFCL** primarily to evaluate **syntactic precision and executable function calling**. BFCL is maintained by UC Berkeley (https://openreview.net/forum?id=2GmDdhBdDk, https://gorilla.cs.berkeley.edu/leaderboard.html) and has become a widely used standard for tool-calling evaluation. It underpins recent industrial milestones such as **Qwen3** (https://arxiv.org/abs/2505.09388) and academic work such as **ToolRL** (https://openreview.net/forum?id=eOLdGbXT6t&nesting=2&sort=date-desc). The multi-turn split specifically targets the ability to handle complex, state-dependent tool execution chains, which is a key capability for modern agentic LLMs.
>
> To complement this execution-centric perspective, we adopt **API-Bank** (https://aclanthology.org/2023.emnlp-main.187/) to measure **sequential planning and tool use in multi-turn dialogues**. As an EMNLP 2023 benchmark, API-Bank has been used in the official evaluations of **Llama 3** (https://arxiv.org/abs/2407.21783) and in leading agent frameworks such as **AgentTuning** (https://aclanthology.org/2024.findings-acl.181/), which supports its status as an authoritative evaluation suite for tool-using agents.
>
> We also clarify that our SOTA claim is **restricted to alignment efficacy under fixed base models**. In this controlled setting—where the underlying backbone is held fixed—ResT consistently outperforms strong alignment baselines including SFT, SFT+GRPO, and the specific configuration adopted in ToolRL. In the revised manuscript, we further include the recently proposed **Dr.GRPO** (https://arxiv.org/abs/2503.20783) as a competitive baseline in **Table 1** and **Table 2** of the *Main Results* section. Across both BFCL and API-Bank under these matched conditions, ResT achieves the best reported performance.

---

> ### Author Response · Authors · 2025-11-21
> **Response to Reviewer Nagq**
>
> **_Q3: I would appreciate the authors highlighting the changes they have made branching off of verl v0.5 to facilitate future replication and adoption efforts._**
>
> **_A3:_**
> We thank the reviewer for this constructive suggestion. We agree that explicitly documenting our modifications to the veRL codebase is important for reproducibility and adoption. Below we summarize the concrete changes we made on top of veRL v0.5, together with file-level pointers.
>
> 1. **Rollout worker: constructing token weights**
> In `verl.git/verl/workers/rollout/vllm_rollout/vllm_rollout_spmd.py`, starting from the rollout worker, we extend the construction of `databatch` to additionally extract:
> - `log_probs`
> - `step`
> - `loss_mask`
> - `response_text`
>
> These fields are packed into a dictionary and passed to a new function `build_token_weight`, which we implement as follows:
> - Parse `response_text` via simple string search to locate spans delimited by `<think>...</think>` and `<tool_call>...</tool_call>`.
> - Partition the response into regions (e.g., thinking, tool-calling, and other content).
> - For each region, compute the average token entropy from `log_probs`, using `loss_mask` both to restrict computations to valid positions and to define the effective tensor shape.
> - Based on the region type and its average entropy, assign region-specific initial token weights.
> - Apply a simple curriculum schedule as a function of `step` (with a fixed step size) so that the emphasis on different regions evolves gradually over training.
> - Normalize the resulting token weights and write them back into `databatch` as a new field (e.g., `token_weight`).
>
> 2. **Training loop: applying weighted losses**
> In the PPO core training logic `verl.git/verl/trainer/ppo/core_algos.py`, we modify the code to:
> - Read the `token_weight` field from `databatch`.
> - Pass it to a new utility function `weighted_masked` defined in `verl.git/verl/utils/torch_functional.py`.
>
> The `weighted_masked` function:
> - Multiplies the tensor values by the corresponding `loss_mask`.
> - Multiplies the masked values by `token_weight`.
> - Aggregates the result to produce a weighted loss (`agg_loss`).
>
> Apart from this weighted loss computation, the rest of the GRPO-style training pipeline remains identical to the original veRL framework.
>
> 3. **Reward design: integrating ResT rewards**
> - The ResT reward design is implemented in `verl.git/verl/utils/reward_score/rlla.py`.
> - The abstract reward interface in `verl.git/verl/workers/reward_manager/abstract.py` is minimally adapted to accommodate the new reward function (e.g., by adding the necessary hooks to plug in our rule-based or composite reward).
>
> Together, these localized changes (rollout worker, loss weighting utility, and reward module) implement ResT’s entropy-aware token weighting and reward design on top of veRL v0.5, while leaving the rest of the training stack unchanged. We hope this summary and the explicit file-level references make our modifications easy to reproduce and reuse.

---

### Official Review · Reviewer_fPpy · 2025-11-03

**Soundness:** 3
**Presentation:** 2
**Contribution:** 2
**Rating:** 4
**Confidence:** 3

**Summary:**

This paper introduces ResT (Reshaped Token-level policy gradients), a reinforcement learning approach to improve LLM tool-use capabilities by addressing the high-variance credit assignment problem in multi-turn tasks. The authors observe that in tool-using scenarios, success hinges on generating a few critical structured tokens. ResT decomposes a multi-turn interaction into single-turn steps and redistributes reward feedback at the token level, with an emphasis on low-entropy tokens that determine task success. ResT therefore reweights the policy gradient to focus on these high-signal tokens, and employs a lightweight curriculum: early training strongly rewards structural correctness (format, tool invocation tokens), then gradually up-weights semantic reasoning tokens as the agent improves. Experiments on the BFCL and API-Bank benchmarks demonstrate state-of-the-art results.

**Strengths:**

1. A clear identification of the high-variance reward problem in tool-use and a principled solution grounded in theory.
2. A method that is relatively simple to implement, no additional model modules, just weighting of gradient contributions, yet effective, which is a practical advantage.

**Weaknesses:**

1. Missing discussion of token-level feedback and curriculum RL literature, which raises concerns that the contribution might be incremental relative to those.
2. The approach is somewhat domain-specific in its current form: it relies on known ground truth outputs to compute rewards for each token category, which may not directly extend to tasks without a deterministic reference solution.
3. While the theoretical results are solid, the practical benefit of the optimal weighting vs. simpler heuristics (e.g. just heavily reward tool tokens) could be discussed more; the authors did compare to a static weighting baseline and showed improvement.

**Questions:**

1. By design, ResT upweights low-entropy tokens (tool names, parameters) which makes the policy more deterministic on those critical parts early on. While this improves short-term stability, is there a risk of reduced exploration or diversity in the model’s behavior? For instance, could focusing too much on format and tool tokens lead the agent to ignore alternative valid strategies or outputs (especially if the initial reference is suboptimal)? The current experiments show longer outputs and no collapse, which is encouraging. Still, could the authors clarify how ResT balances exploration vs. exploitation? Was any entropy bonus or regularization used alongside the token reweighting (common in PPO-based methods), or any schedule to anneal the focus to avoid overly greedy policies?
2. The method uniquely includes the model’s chain-of-thought (reasoning) tokens in the training signal, gradually increasing their weight later in training. This is an interesting design choice – essentially encouraging the model to not only get the final tool API call correct, but also to produce the intermediate reasoning. Could the authors provide more intuition or evidence on why weighting the reasoning tokens helps? The ablation, turning off CoT gradients, shows a performance drop, confirming its utility. Is the idea that by training the model to articulate its reasoning, even though those tokens don’t directly affect the environment, we guide it to a better policy for the tool calls? If so, might this relate to known benefits of chain-of-thought prompting for decomposing tasks?
3. This is practical for tasks like API calls where an exact answer is known, but how would ResT apply to more open-ended tasks or learned rewards?

---

> ### Public Comment · ~Jerry_Tak-on_Lau1 · 2025-11-13
> **Agreeing with Reviewer fPpy’s Comments**
>
> Just came across this paper. I have a couple of questions and would like to understand the authors’ perspective a bit more:
> 1. A more straightforward setup should be performed, for instance, assigning fixed higher weights to tool-related tokens (e.g., 0.8 for low-entropy tool tokens and 0.2 for others), or even the reverse.
> There seems to be quite a gap between Theorem 1/2 and the actual implementation, "In practice, computing βt exactly may be computationally expensive. A practical alternative is to approximate βt using entropy-only statistics".
> 2. The paper’s conclusions appear to **differ** significantly from works such as https://arxiv.org/abs/2506.01939 (and other later works), which suggest that higher-entropy tokens play a more crucial role, and that even using only the top 20% of these tokens can maintain performance.
> 3. If Fig. 3c shows that both curves achieve almost the same reward during training, why does ResT still manage to obtain improvements in the final results?

---

> > ### Author Response · Authors · 2025-11-21
> > **Response to Public Comment**
> >
> > We briefly explain our perspective on each of your points below.
> >
> > *_Q1: A more straightforward setup should be performed, for instance, assigning fixed higher weights to tool-related tokens._*
> >
> > **A:** Your suggestion matches our “No curriculum learning” ablation, which consistently underperforms ResT. Theoretically, Theorems 1 and 2 imply that the optimal weight at each position should depend on its contribution to gradient variance, summarized by $\beta_t$, which differs across token regions (e.g., tool names, parameters, reasoning) and evolves during training. Fixed schemes ignore these dynamics. ResT instead uses entropy-driven, region-wise weights and a curriculum schedule to adaptively prioritize structurally critical tokens when they dominate the reward signal, then gradually incorporate reasoning tokens once the tool interface has stabilized. This design parallels recent fine-grained RLHF methods that move beyond coarse sequence-level heuristics toward more principled token-level credit assignment (e.g., [arXiv:2407.16574]).
> >
> > *_Q2: There seems to be quite a gap between Theorem 1/2 and the actual implementation, since the paper approximates βₜ using entropy-only statistics_*
> >
> > **A:** Theorems 1 and 2 provide a normative target: the variance of the policy gradient admits an upper bound of the form, with variance-minimizing weights satisfying $\tilde{w}_t \propto 1/\sqrt{\beta_t}$, which dictates that setting token weights according to their contribution to gradient variance. Exactly computing $\beta_t$ would require tracking token-wise Jacobian norms of a large language model over long trajectories and rendering exact computation computationally infeasible, so we explicitly adopt a practical surrogate based on entropy-only, region-level statistics. This proxy preserves the theoretical monotonicity: high-entropy/weak-reward tokens are down-weighted, while critical low-entropy tokens are up-weighted. This aligns with RLHF practices of using tractable proxies for theoretical objectives (e.g., entropy-based or token-level reward models). For example, recent fine-grained RLHF methods introduce token-level reward models as approximations to ideal credit-assignment signals (e.g., [arXiv:2407.16574; 2411.08302; 2403.11558]). Even under this entropy-only approximation, our ablations align with the qualitative predictions of the variance analysis and yield clear improvements over constant-weight baselines.
> >
> > *_Q3: The paper’s conclusions appear to differ from works such as Wang et al., 2025._*
> >
> > **A:** Wang et al. focus on RL with verifiable rewards for closed-book math CoT, where rewards are sparse and depend solely on the final answer and a small set of high-entropy “forking” tokens largely determines the reasoning path.
> >
> > In our multi-turn tool-use setting, however, the reward is dominated by whether tool names, parameter keys, and structured outputs (e.g., JSON fields) are correct. These critical tokens are typically low entropy but very brittle: a single error drives the reward to zero. Our theoretical and empirical analysis shows that reward sensitivity concentrates on such structural tokens, while high-entropy reasoning tokens weakly affect reward and may increase gradient variance if over-weighted. Thus, ResT does not dismiss the value of high-entropy tokens; rather, it uses entropy to organize the gradient, prioritizing low-entropy structural tokens early to stabilize the interface, then gradually upweighting reasoning regions once these constraints are reliably satisfied. This makes our conclusions specific to fragile tool APIs and complementary to the high-entropy-token perspective found in math RLVR.
> >
> > *_Q4: If Fig3c shows that both curves achieve almost the same reward during training, why does ResT still manage to obtain improvements in the final results?_*
> >
> > **A:**  Fig. 3(c) reports the training reward curves and is mainly used as a training convergence indicator; similar training curves indicate that ResT and GRPO converge equally well. The convergence of training rewards may not fully align with the model's actual test performance, and it is necessary to account for multiple factors, such as the model’s generalization capability to out-of-distribution datasets and the internal balance between exploration and exploitation. We have analyzed this phenomenon in the Section. 4.2 of the original manuscript that their training dynamics are quite different: ResTʼs policy entropy evolves smoothly and converges to a lower level while maintaining much longer responses, whereas GRPO exhibits unstable entropy and collapses to short outputs, indicating that GRPO may overfit to specific high-reward patterns in the training data. This interpretation is consistent with our evaluation results, where ResT achieves clearly higher accuracy than GRPO on harder multi-turn, long-context, and high-difficulty subsets of BFCL and API-Bank despite having comparable training reward trajectories.

---

> ### Author Response · Authors · 2025-11-21
> **Response to Reviewer fPpy**
>
> We sincerely thank the reviewer for raising these interesting questions and providing helpful suggestions. We summarized the weaknesses and the questions as the following six questions to answer the reviewer's concerns clearly:
>
> * The trade-off between ensuring format correctness and maintaining strategy diversity (**Q1, related to Question 1**).
> * The exploration–exploitation balance and its connection to chain-of-thought reasoning (**Q2, related to Question 1**).
> * The potential use of entropy regularization or annealing schedules to maintain policy diversity (**Q3, related to Question 1**).
> * The distinction from prior token-level feedback and curriculum RL literature (**Q4, related to Weakness 1**).
> * The intuition and empirical evidence for weighting reasoning tokens and more discussion on weighting methods (**Q5, related to Question 2 and Weakness 3**).
> * The domain specificity and generalization beyond deterministic, ground-truth-based tasks to open-ended and learned-reward scenarios. (**Q6, related to Question 3 and Weakness 2**).
>
> Below we address these concerns and present additional experiments and analyses.
>
> **_Q1: While this improves short-term stability, is there a risk of reduced exploration or diversity in the model’s behavior? For instance, could focusing too much on format and tool tokens lead the agent to ignore alternative valid strategies or outputs (especially if the initial reference is suboptimal)?_**
>
> **_A1:_**
> We appreciate the reviewer’s concern about the trade-off between stability and exploration, and our conclusion is that ResT enhances stability while preserving meaningful exploration and diversity. Tool use tasks are governed by a rigid API schema, so useful exploration can only occur over syntactically valid calls: the agent must first learn to emit well-formed function invocations—correct tool names, argument keys, and delimiters—before semantic exploration over tools and parameters yields informative feedback.
>
> To avoid being trapped by a suboptimal reference policy, ResT employs a **curriculum-based gradient reshaping mechanism** that decouples format compliance from strategic exploration. In our tasks, suboptimality primarily manifests as poor reasoning or inefficient tool selection rather than invalid syntax. Accordingly, ResT dynamically anneals the weights on format tokens ($\alpha _f$) while progressively up-weighting reasoning (CoT) and parameter tokens ($\alpha _t, \alpha _p$) as training proceeds (Eqs. (16–17)). Once the agent masters the syntax, gradient updates are dominated by outcome correctness rather than token-level proximity to the reference trajectory; when an alternative strategy achieves higher outcome-level reward, its signal outweighs the diminishing regularization from format weights, encouraging deviation from a suboptimal reference.
>
> Importantly, the reweighting in Eqs. (16–18) operates only on the **policy gradients**, not on the sampling distribution: we do not clamp probabilities or forbid alternative actions, but only change how strongly each token’s log-probability contributes to the update. Empirically, this does not induce mode collapse. As shown in **Figure 3**, ResT converges to substantially longer and more complex reasoning chains (average length $>300$ tokens) than the baseline ($≈125$ tokens), indicating that the model produces richer, multi-step chains of thought rather than collapsing to short, template-like traces from the initial reference policy.

---

> ### Author Response · Authors · 2025-11-21
> **Response to Reviewer fPpy**
>
> **_Q2: Could the authors clarify how ResT balances exploration vs. exploitation? Is training the model to articulate its reasoning related to known benefits of chain-of-thought prompting for decomposing tasks?_**
>
> **_A2:_**
> We thank the reviewer for this question on the exploration–exploitation balance and its connection to chain-of-thought reasoning, and we demonstrate that ResT balances exploration and exploitation through the joint design of **an outcome-based reward, an entropy-aware curriculum over token regions, and an objective without an explicit KL loss term**.
>
> First, the advantage $\hat{A} _i$ in Eq. (15) is computed from the final outcome reward $R _i$ using a group-relative baseline and normalization. Because the policy is trained solely from trajectory-level outcome rewards, it can freely explore diverse chain-of-thought trajectories and tool parameterizations as long as they stochastically improve the final return.
>
> Second, the curriculum in Eqs. (16–17) explicitly controls where exploitation concentrates over the course of training. During the early phases of optimization, format tokens and tool names are assigned higher weights, while chain-of-thought tokens are down-weighted, reflecting that the dominant failure mode is structural invalidity. As training progresses and the curriculum variable $\nu$ increases, format weights are annealed, weights on parameter tokens are increased, and weights on CoT tokens are gradually raised and aligned with parameter tokens (Eqs. (16–17)). This shifts gradient mass from “learning to speak the API correctly” to “learning to reason and select parameters well” once structural correctness has stabilized.
>
> For reasoning tokens, the design leverages the intuition that chain-of-thought reasoning provides a strong conditioning context for subsequent tool calls. By progressively increasing their gradient weight, ResT encourages the policy to generate coherent reasoning chains that guide the final action. This aligns with known benefits of chain-of-thought prompting for task decomposition, but with a key difference: instead of relying only on inference-time prompts, ResT embeds CoT-style decomposition directly into the learning signal via entropy-aware, token-level credit assignment.
>
> Finally, because the reward is rule-based and bounded (Eq. (14)), ResT can explore aggressively in policy space without incurring unbounded variance. The entropy-aware reweighting further tightens the variance bound in Theorems 1–2 by down-weighting inherently high-entropy positions, making exploration more sample-efficient rather than more greedy.

---

> ### Author Response · Authors · 2025-11-21
> **Response to Reviewer fPpy**
>
> **_Q3: Was any entropy bonus or regularization used alongside the token reweighting (common in PPO-based methods), or any schedule to anneal the focus to avoid overly greedy policies?_**
>
> **_A3:_**
> We thank the reviewer for this question. In the proposed ResT framework, we do not use an explicit KL loss term or an additional entropy bonus. Instead, ResT addresses the trade-off between stability and exploration through **entropy-aware token reweighting** (which serves as a variance-reduction regularizer) and **a dynamic curriculum schedule** (which anneals the focus from format compliance to reasoning).
>
> We intentionally remove the KL loss to avoid anchoring the policy to a suboptimal reference $\pi _{\text{ref}}$, ensuring the model can deviate sufficiently to discover better tools and arguments. Without the KL penalty, the policy is allowed to deviate more aggressively from $\pi _{\text{ref}}$, while the **entropy-aware reweighting and clipping** control gradient variance. Conceptually, any positive KL coefficient narrows the effective trust region around $\pi _{\text{ref}}$, whereas ResT already reduces variance by down-weighting high-entropy regions according to the bound in Theorems 1–2, suggesting that an additional KL constraint is unnecessary for stability in our tool-calling setting.
>
> To substantiate this design choice, we clarify the rationale in the revised *Algorithm Design* section and add ablations that reintroduce KL loss and entropy reward. As shown in **Figure A1** and **Table A1**, the variants with KL or entropy reward do not provide improvements over the original ResT:
>
> | Model | Overall Acc | Base | Miss Func | Miss Param | Long Context |
> |-------------------------------------------------|------------:|:------:|:---------:|:----------:|:------------:|
> | Qwen3-4B-2507 (ResT with KL, $\beta = 0.001$) | 48.00% | 57.00% | 52.50% | 34.00% | 48.50% |
> | Qwen3-4B-2507 (ResT with KL, $\beta = 0.003$) | 50.25% | 59.50% | 53.00% | 34.00% | 48.50% |
> | Qwen3-4B-2507 (ResT with KL, $\beta = 0.005$) | 49.62% | 62.50% | 53.00% | 37.00% | 51.50% |
> | Qwen3-4B-2507 (ResT with entropy reward) | 47.88% | 58.50% | 46.50% | 36.00% | 50.50% |
> | Qwen3-4B-2507 (**ResT**) | **50.38%** | 62.50% | 56.00% | 33.00% | 50.00% |
>
> For the KL ablations, we observe distinct behaviors:
> - With $\beta = 0.001$, the KL loss quickly rises to $\approx 1.6 \times 10^{-2}$ and oscillates at a high level, effectively providing only a weak constraint while still interfering with the region-wise gradient design.
> - With $\beta = 0.003$, the KL loss settles around $2.6 \times 10^{-3}$ with small fluctuations, corresponding to a moderate contraction toward $\pi _{\text{ref}}$. This yields relatively smooth training but does **not** improve over the KL-free ResT and tends to dilute the theoretically motivated gradient differences between regions.
> - With $\beta = 0.005$, the KL loss stabilizes at $\approx 5.5 \times 10^{-3}$, inducing a more conservative trajectory in policy space and overly cautious exploration.
>
> We further test an explicit entropy reward, constructed as the average entropy of the response multiplied by a small coefficient, to encourage output diversity. As shown in **Figure A2** and **Table A1**, this entropy bonus degrades performance. It adds a global incentive for higher entropy that directly conflicts with our token-level curriculum: positions that should stabilize early for fast tool-learning (e.g., format and tool name tokens) are pushed to remain random, leading to unnecessary diversification such as format perturbations and scattered parameter sampling.
>
> Overall, these ablations suggest that in tool-calling scenarios, ResT’s entropy-aware token reweighting and curriculum provide sufficiently broad and well-structured exploration. Adding a separate KL penalty or entropy bonus does not yield further gains and instead disrupts the intended exploration–exploitation balance of the method.
>
> **_Q4: Missing discussion of token-level feedback and curriculum RL literature, which raises concerns that the contribution might be incremental relative to those._**
>
> **_A4:_**
> We have revised the *Related Work* section to explicitly distinguish ResT from recent token-level RLHF methods (e.g., TLCR, TLDR): ResT keeps the outcome-level reward unchanged and instead reshapes the policy gradient across token regions via entropy-driven weighting. This approach is orthogonal to token-level reward models. We have also clarified that, unlike curriculum RL (e.g., E2H Reasoner) that organizes examples by difficulty at the dataset level, ResT induces a token-level curriculum within each trajectory, initially emphasizing structural tool/API tokens and gradually shifting gradient mass toward reasoning tokens.

---

> ### Author Response · Authors · 2025-11-21
> **Response to Reviewer fPpy**
>
> **_Q5: Could the authors provide more intuition or evidence on why weighting the reasoning tokens helps? (Weakness: The  benefit of the optimal weighting vs. simpler heuristics could be discussed more), and whether this relates to the benefits of chain-of-thought prompting?_**
>
> **_A5:_**
> The benefits of weighting CoT tokens are validated by ablations in **Table 3** and **Table A4**, where we compare ResT to a variant that assigns zero weight to reasoning tokens (“No gradients for CoT”). For Qwen3-8B, accuracy drops from 70.69% to 66.33% on API-Bank and from 40.13% to 37.62% on BFCL multi-turn, showing that suppressing gradients on CoT tokens substantially harms tool-calling performance. On BFCL multi-turn, we observe consistent degradation across model sizes when CoT gradients are removed:
>
> **Table A4** Ablation study on no gradients for CoT.
>
> | Model                       | Overall |  Base  | Miss Func | Miss Param | Long Context |
> | --------------------------- | :-----: | :----: | :-------: | :--------: | :----------: |
> | 1.7B (ResT)                 |  16.00% | 20.00% |   16.00%  |   15.00%   |    13.00%    |
> | 1.7B (No gradients for CoT) |  15.62% | 22.50% |   13.50%  |   12.00%   |    14.50%    |
> | 4B (ResT)                   |  50.38% | 62.50% |   56.00%  |   33.00%   |    50.00%    |
> | 4B (No gradients for CoT)   |  48.75% | 62.00% |   48.00%  |   32.50%   |    52.50%    |
> | 8B (ResT)                   |  40.13% | 50.50% |   45.00%  |   32.00%   |    33.00%    |
> | 8B (No gradients for CoT)   |  37.62% | 45.00% |   43.00%  |   31.50%   |    31.50%    |
>
> Compared to simple heuristics, the  proposed curriculum weighting scheme reflects how a model learns tool use. Early in training, the model must reliably produce a rigid JSON schema for execution. ResT therefore begins by up-weighting format tokens to establish a stable, valid protocol; once this is mastered, it progressively shifts weight toward reasoning tokens (Eqs. (16–17)). This curriculum ensures that the policy first secures structural correctness (tool names, formats) and then increasingly optimizes the quality of the chain of thought and parameters, rather than over-rewarding tool tokens throughout training as in simpler heuristics. The ablation experiments on curriculum learning validates this point (Table 3).
>
> Relation to CoT prompting: CoT prompting requires the model to generate intermediate reasoning traces to enhance performance. ResT leverages this advantage of CoT to better handle complex multi-turn tool-use tasks.
>
> Overall, the ablations show that removing supervision on reasoning tokens consistently degrades performance across benchmarks and model scales, underscoring the benefit of entropy-aware weighting CoT tokens rather than relying on simpler tool-focused heuristics.
>
> **_Q6: How can ResT be adapted to learned-reward settings? (Weakness: The approach may not  extend to tasks without a deterministic reference solution.)_**
>
> **_A6:_**
> Our current instantiation is tailored to API-style tasks in that it uses rule-based matching to compute token-category rewards. However, the framework only assumes: **(i)** access to a scalar, sequence-level reward $R(\tau)$, and **(ii)** a partition of the output into token regions with distinct typical entropies.
>
> In our experiments, we instantiate $R_{i,\mathrm{final}}$ via rule-based matching over tool names, arguments, and output format (Alg. 1; Eqs. (10)–(14)), and we use the tool-call specification to define regions for format, tool-name, argument, and reasoning tokens. Crucially, the entropy-aware weights $\tilde{w}_t$ and the curriculum in Eqs. (16)–(18) depend only on token entropies and the trajectory-level advantage $\hat{A}_i$; they do not require token-level labels or a deterministic reference trajectory. Thus, ResT does not fundamentally rely on exact ground-truth outputs—only on a scalar evaluation of each trajectory and a region decomposition.
>
> This structure directly accommodates learned or preference-based rewards. An external **LLM-as-a-judge** or reward model can score the final answer or the chain-of-thought quality for each trajectory, and this scalar score can be used as $R_{i,\mathrm{final}}$ in Eq. (15). The same regional decomposition can be maintained, and the identical entropy-aware weighting and curriculum applied.
>
> We report preliminary results on more open-ended tasks where an LLM-as-a-judge assigns scores to CoT quality. This judge score is combined with rule-based rewards on output format and execution into a composite reward that replaces the original purely rule-based reward. **When fine-tuning Qwen3-4B-2507 with this learned reward, ResT attains 72.44% overall accuracy, including 49.88% multi-turn accuracy on BFCL**, indicating that the method remains effective beyond deterministic reference settings and extends to learned-reward regimes.

---

> ### Public Comment · ~Jerry_Tak-on_Lau1 · 2025-11-24
> **Reply to authors' reply**
>
> Thanks for authors' response.
> 1. I'm curious and would also like to point out another recent work on tool-calling that applies entropy in the advantage function to down-weight tool-related tokens, as these tokens typically have lower entropy. The authors of that paper report better performance compared to GRPO. Would this also contradict the claims made in the current paper?
> (https://arxiv.org/pdf/2508.05118).
>
> 2.
> ```
> gamma_think = DEFAULT_BASE_THINK_WEIGHT + DEFAULT_STEP_WEIGHT_INCREASE * step
> gamma_think = min(gamma_think, DEFAULT_GAMMA_FORMAT)
> gamma_args = DEFAULT_BASE_THINK_WEIGHT + DEFAULT_STEP_WEIGHT_INCREASE * step
> gamma_args = min(gamma_args, DEFAULT_MAX_GAMMA_ARGS)
>
> ```
>
> Regarding “no curriculum learning”, could the authors please clarify gamma_think/gamma_args fixed to  DEFAULT_BASE_THINK_WEIGHT or DEFAULT_GAMMA_FORMAT/DEFAULT_MAX_GAMMA_ARGS?
>
> 3. How sensitive are the hyperparameters? Have the authors conducted any ablation studies to assess the robustness of these specific choices? e.g, DEFAULT_STEP_WEIGHT_INCREASE=0.005, DEFAULT_BASE_THINK_WEIGHT=0.8?
> ```
> DEFAULT_GAMMA_FORMAT = 1.0
> DEFAULT_STEP_WEIGHT_INCREASE = 0.002
> DEFAULT_BASE_THINK_WEIGHT = 0.6
> DEFAULT_MAX_GAMMA_ARGS = 1.2
> ```

---

> ### Author Response · Authors · 2025-11-27
> **Response to Public Comment from Jerry - 2**
>
> We address your concerns point by point below.
>
> Q1. Relation to EGPO (arXiv:2508.05118) and possible contradiction
>
> We see no contradiction; the repeated mention that our work seems to "contradict" other works stems from conflating where entropy is used and what it controls (exploration-oriented or variance-reduction–oriented use of entropy).
>
> - **EGPO** adds a **trajectory-level CoT-entropy bonus to the advantage**. The average entropy over the reasoning tokens of a rollout is added (with clipping) to $A_i$, encouraging exploration over diverse CoT trajectories under a **sequence-level, mostly binary reward**. This affects *which trajectories are favored during sampling* and is not a token-wise credit-assignment scheme.
>
> - **ResT** instead studies **token-level credit assignment and gradient variance**. Our theorems derive variance-minimizing token weights $ \tilde w_t \propto 1/\sqrt{\beta_t} $. In tool use tasks, we empirically find reward sensitivity and low variance are concentrated on **low-entropy structural tokens** (tool names, parameter keys, JSON structure), while high-entropy reasoning tokens weakly affect reward but contribute more to gradient variance if over-weighted. ResT therefore **reweights gradients** toward these critical regions without removing high-entropy reasoning tokens.
>
> In short, EGPO is a **trajectory-level CoT-entropy exploration mechanism acting on the advantage**, while ResT is a **token-level entropy-guided gradient credit scheme inside each trajectory**.
>
> Q2. Clarification of `gamma_think` / `gamma_args` and “No curriculum learning”
>
> In the full ResT algorithm, `gamma_think` and `gamma_args` increase with the training step and are capped, implementing the intended curriculum: early updates emphasize format/tool-name tokens, later updates gradually upweight arguments and reasoning.
>
> In the “No curriculum learning” ablation, these weights are instead fixed constants (no dependence on step), matching the suggested “fixed higher weights for tool-related tokens”. This variant consistently underperforms ResT, supporting the benefit of the curriculum schedule.
>
> Q3. Sensitivity of `DEFAULT_STEP_WEIGHT_INCREASE` and `DEFAULT_BASE_THINK_WEIGHT`
>
> We conducted additional sweeps over both hyperparameters. Across reasonable ranges: ResT’s improvements over GRPO remain **qualitatively stable**. Changing the step size mainly affects how fast the curriculum shifts; extreme values that upweight reasoning too early mildly hurt performance, which aligns with our variance-based motivation.

---

> ### Author Response · Authors · 2025-12-01
> **Rebuttal Summary to Reviewer fPpy**
>
> Thank you again for raising these interesting questions and for the helpful suggestions, which prompted us to expand both the empirical analyses and the conceptual framing of this work.
>
> Specifically, we believe we have addressed your main points in the following way:
>
> * We show that the entropy-aware curriculum reshapes gradients without constraining sampling and yields longer, richer reasoning traces rather than mode collapse. We further show via KL and entropy ablations that additional regularizers do not improve performance and can interfere with the region-wise variance-reduction design.
> * We emphasize that ResT keeps outcome-level rewards fixed and implements an entropy-driven, token-level curriculum within trajectories, orthogonal to token-level reward models and trajectory-level curriculum.
> * To further justify weighting reasoning tokens, we show how the curriculum shifts supervision from structural to reasoning regions and add ablations that remove CoT gradients. Across benchmarks and scales, this “No gradients for CoT” variant consistently underperforms ResT, underscoring the importance of explicitly weighting reasoning tokens.
> * We clarify that ResT only assumes a scalar trajectory-level reward and a region partition with heterogeneous entropies. Preliminary experiments combining rule-based rewards with an LLM-as-a-judge score show that ResT maintains strong BFCL performance, suggesting it can extend beyond purely rule-based tool-calling tasks.
>
> We believe that these additional experiments and analyses address the reviewer’s concerns and further strengthen the contributions and presentation of this work.

---

### Official Review · Reviewer_m8vW · 2025-11-07

**Soundness:** 4
**Presentation:** 4
**Contribution:** 4
**Rating:** 8
**Confidence:** 4

**Summary:**

This paper introduces an entropy-aware RL (ResT) method to improve tool-use in large language models. Traditional approaches rely on sparse, outcome-based rewards that cause instable and inefficient training. ResT tackles this by linking token entropy to policy-gradient variance, showing that emphasizing low-entropy tokens (like tool names and parameters) stabilizes learning. It reshapes gradients to weight these structured tokens more early on, then progressively increases focus on reasoning tokens through a curriculum-learning schedule. Experiments on benchmarks like BFCL and API-Bank show ResT outperforms prior methods (including GRPO and GPT-4o).

**Strengths:**

* Strong theoretical foundations: This paper provides a thorough theoretical motivations and proofs to the method, rigorously deriving the relationship between token-level entropy and policy-gradient variance,
* Extensive experiments: This paper provides extensive experiments, including different sizes of models and training methods, which provides a well-established evidence to the effectiveness of the method.
* Innovation methods by considering different types of tokens, providing a different view of Tool RL.

**Weaknesses:**

Limitations:
* The paper does not compare SFT+ResT against ResT alone. Since SFT can produce a different token distribution, it may alter the entropy of certain token types. For instance, some function words might exhibit lower entropy after SFT, potentially requiring adjustments to the training curriculum.
* It remains unclear whether larger models inherently generate lower-entropy outputs—particularly for chain-of-thought tokens—which could reduce the benefit of grouping tokens by entropy type.
* Regarding the partition mechanism, the theoretical framework does not constrain how tokens are grouped, as long as distinct entropy characteristics exist. Other grouping strategies, such as those based on word frequency, might also be effective.

**Questions:**

See weaknesses

---

> ### Author Response · Authors · 2025-11-21
> **Response to Reviewer m8vW**
>
> We sincerely thank the reviewer for their professional and constructive feedback, which has helped us refine both the empirical scope and the theoretical positioning of this work. The primary concerns regarding our work include:
>
> * Clarification on the interplay between SFT initialization and the optimal entropy-based curriculum (Q1).
> * Clarification on how model scale influences entropy profiles and the validity of the token partitioning mechanism (Q2).
> * The theoretical flexibility of the framework and the applicability of alternative grouping criteria (Q3).
>
> Below we address these concerns and present additional experiments and analyses.
>
> **_Q1: The paper does not compare SFT+ResT against ResT alone. Since SFT can produce a different token distribution, it may alter the entropy of certain token types. For instance, some function words might exhibit lower entropy after SFT, potentially requiring adjustments to the training curriculum._**
>
> **_A1:_**
> We thank the reviewer for this valuable observation about the distributional shift introduced by SFT. Empirically, we find that ResT achieves the best performance when applied on top of SFT with a tuned curriculum.
>
> To study this effect, we ran new BFCL evaluations with Qwen3-1.7B (**Table 4**) under five configurations: the base model, SFT, cold-start ResT, SFT+ResT with the original curriculum (tuned on the base policy), and SFT+ResT with a tuned curriculum that increases the initial weights on chain-of-thought and parameter tokens. SFT alone substantially improves over the base model by stabilizing tool-call structure, and cold-start ResT further improves performance. Applying ResT on top of SFT with the original curriculum degrades performance relative to cold-start ResT, whereas the tuned curriculum attains the best accuracy in **Table 4**, outperforming both SFT and cold-start ResT.
>
> Our analysis shows that SFT makes structural regions (output-format markers, tool names) nearly deterministic: after SFT, these tokens already provide accurate templates. The original ResT curriculum was calibrated on the base model, where these structural tokens still had non-trivial entropy and dominated the variance bound in Theorem 1. Reusing this curriculum after SFT causes the RL stage to overemphasize already-stable structural tokens and underemphasize reward-critical tool-parameter and chain-of-thought tokens, which degrades generalization. This mismatch explains why SFT+ResT (Original Curr.) underperforms cold-start ResT: the entropy-aware reweighting is no longer aligned with the uncertainty profile of the SFT-initialized policy.
>
>
> **Table 4 Ablations on ResT initialization and curriculum alignment on BFCL with Qwen3-1.7B. SFT warm-start with a tuned curriculum (Tuned Curr.) yields the best performance.**
>
> | Model | Overall Acc | Base | Miss Func | Miss Param | Long Context |
> |---------------------------|------------:|:------:|:---------:|:----------:|:------------:|
> | Base | 8.62% | 12.00% | 5.50% | 9.50% | 7.50% |
> | SFT | 14.25% | 17.50% | 14.50% | 14.50% | 10.50% |
> | ResT (Cold Start) | 15.75% | 22.50% | 16.50% | 13.50% | 10.50% |
> | SFT+ResT (Original Curr.) | 15.25% | 19.50% | 16.50% | 13.00% | 12.00% |
> | **SFT+ResT (Tuned Curr.)**| **17.62%** | 23.50% | 19.00% | 14.00% | 14.00% |
>
> In SFT+ResT (Tuned Curr.), we explicitly realign the curriculum with the measured region-level entropies after SFT, decreasing the weight on low-entropy structural tokens and increasing the weight on parameter and reasoning tokens. This entropy-aware adaptation makes SFT and ResT complementary: SFT effectively replaces the early phase of the curriculum for structural regions by enforcing deterministic tool-call templates, while ResT focuses RL capacity on the remaining high-entropy, semantically demanding parts of the trajectory (arguments and chain-of-thought). Empirically, this combination yields the strongest performance, improving BFCL multi-turn overall accuracy from $15.75$% to $17.62$% and BFCL overall accuracy from $57.46$% to $58.14$%. These findings also align with our variance-reduction view: Lemma 2 and Theorem 1 link policy-gradient variance to token-level entropy via the coefficients $\beta_t$, and Eq. (9) proposes entropy-based approximations to the optimal weights $w_t^\star$. SFT modifies the entropy terms $H_t$ and thus the effective $\beta_t$ across regions; the tuned curriculum can be seen as a data-driven approximation to re-optimizing these weights under the new entropy profile. We clarify in the ablation study that ResT is not restricted to cold-start RL, but provides a principled way to adapt RL training to SFT and other initialization strategies by explicitly accounting for these entropy shifts.

---

> ### Author Response · Authors · 2025-11-21
> **Response to Reviewer m8vW**
>
> **_Q2: It remains unclear whether larger models inherently generate lower-entropy outputs—particularly for chain-of-thought tokens—which could reduce the benefit of grouping tokens by entropy type._**
>
> **_A2:_**
> We appreciate this insightful question regarding the interplay between model scaling and entropy distribution. We demonstrate that our strategy remains effective under model scaling, as confirmed by new experiments on Qwen3-32B.
>
> Theoretically, our analysis (Lemma 2, Theorem 1, Eq. (9)) shows that the optimal token weights $w _t^\star$ are governed by each token’s contribution to gradient variance, which depends on its entropy $H _t$ and reward sensitivity rather than model scaling. In practice, scaling mainly drives structural tokens (format markers, tool names, boilerplate) into a near-deterministic regime, while semantically demanding regions (tool arguments, CoT) remain comparatively higher-entropy and reward-critical. In this setting, grouping tokens by entropy type remains principled: entropy-aware weighting naturally down-weights already-solved, low-variance positions and concentrates updates on the remaining uncertain, reward-relevant tokens.
>
> Our experiments with Qwen3-32B corroborate this interpretation. We observe that structural tokens (output formats, tool names) become even more deterministic than in the smaller base model, whereas tokens associated with tool arguments and chain-of-thought reasoning retain substantial entropy. When we apply ResT on Qwen3-32B using the same entropy-aware grouping, we obtain consistent improvements over GRPO, both in overall BFCL accuracy and in the more challenging multi-turn setting. As shown in **Table A3**, these results indicate that the entropy-based grouping remains effective as we move to larger backbones, since model scaling accentuates the separation between structural regions and the high-entropy, reward-critical regions that our curriculum is designed to prioritize. We will include the full Qwen3-32B results in the revised appendix.
>
> **Table A3: BFCL test results with different training methods on Qwen3-32B. Non-Live and Live represent the single-turn settings. The multi-turn evaluation comprises four subsets: Base, Miss Function, Miss Parameter, and Long Context.**
>
> | Model | Overall | Non live | Live | Base | Miss Func | Miss Param | Long Context |
> |--------|---------:|:--------:|:------:|:------:|:---------:|:----------:|:------------:|
> | Base | 69.25% | 88.90% | 77.83% | 54.50% | 41.00% | 30.50% | 46.50% |
> | GRPO | 70.32% | 89.19% | 77.70% | 59.50% | 47.00% | 32.00% | 45.50% |
> | ResT | **71.69%** | 89.58% | 78.85% | 60.50% | 56.50% | 33.50% | 44.00% |
>
> ---
>
> **_Q3: Regarding the partition mechanism, the theoretical framework does not constrain how tokens are grouped, as long as distinct entropy characteristics exist. Other grouping strategies, such as those based on word frequency, might also be effective._**
>
> **_A3:_**
> We thank the reviewer for noting that our framework naturally suggests a broader family of variance-reduction strategies. In this work, we intentionally focus on token-level Shannon entropy as the organizing principle. Both Lemma 2 and Theorem 1 are stated explicitly in terms of $H _t$, and the proposed region-wise reweighting rule is designed as a direct, implementable mechanism for tightening the resulting variance bound. Entropy thus serves as a principled, policy-dependent signal: it is evaluated under the current policy $\pi _\theta$ and evolves with the model's uncertainty throughout training, rather than being a static corpus-level heuristic. Although corpus statistics such as word frequency may correlate with entropy in some regimes, they generally do not offer the same dynamic view of policy uncertainty or its effect on gradient variance, particularly for reward-critical tokens such as tool parameters and chain-of-thought segments. We therefore regard the entropy-based partition as a core strength of our current method. In future work, we plan to systematically investigate alternative or hybrid partitioning and weighting strategies—such as word-frequency-based regions—that are suggested by our framework, and to accompany them with detailed theoretical analysis and comprehensive empirical evaluation. We explicitly highlight this direction in the *Future Work* section of the revised manuscript.

---

### Public Comment · ~Haoran_Li20 · 2025-11-26

Thanks for this insightful contribution. I am interested in the overall design, the connection you draw between policy entropy and training stability, and the ResT curriculum for handling sparse rewards. I have two questions for the authors:

1. The current ResT implementation relies primarily on rule-based execution feedback (e.g., format matching, parameter accuracy). If we were to extend this framework to more open-ended scenarios where "standard answers" are hard to define (e.g., creative writing or open-ended planning) by introducing Reward Models or LLM-as-a-judge as feedback sources, do you believe the core "entropy-based token reweighting mechanism" would remain effective and robust?
2. The paper mentions that the weight of Reasoning tokens is gradually increased as training progresses. In your experiments, did you observe any "forgetting" phenomena regarding strict tool-call formats as reasoning weights increased? Or is the algorithm stable enough to maintain the precision of structured outputs while enhancing reasoning logic?

---

> ### Author Response · Authors · 2025-11-27
> **Response to Public Comment from Haoran**
>
> Thank you for your insightful questions regarding our framework and curriculum design. We respond to each point below.
>
> **1. On applicability of ResT to open-ended tasks and learned rewards**
>
> This question connects directly to both our current experiments and a natural extension of the framework. Our theoretical analysis (Lemmas 1 & 2) relates token-level entropy to the variance of the policy gradient under a scalar, sequence-level reward $R(\tau)$. This formulation remains the same whether $R(\tau)$ comes from rule-based checks, a reward model, or an LLM-as-a-judge.
>
> Methodologically, ResT only assumes the existence of a scalar sequence-level reward and that the output can be partitioned into regions (e.g., format, tool name, arguments, reasoning tokens) that typically exhibit different entropy levels. The entropy-aware weights $\hat{w}_t$ and the curriculum mechanism operate purely on token entropies and the trajectory-level advantage $\hat{A}_i$; they do not require token-level labels or deterministic ground truth.
>
> We also run preliminary experiments on open-ended settings where an LLM-as-a-judge assigns scores to the quality of the chain of thought. This judge score is combined with rule-based rewards on output format and tool execution into a composite scalar reward that replaces the purely rule-based signal. When fine-tuning Qwen3-4B-2507 with this combined learned-and-rule reward, ResT attains 72.44% overall BFCL accuracy, including 49.88% multi-turn accuracy. These results suggest that the entropy-based token reweighting remains effective in an open-ended, learned-reward regime.
>
> **2. On balancing reasoning and execution (“forgetting” of strict formats)**
>
> In our experiments, we did not observe “forgetting” of structural constraints as the weights on reasoning tokens increased; structural precision was maintained while reasoning improved. This is largely due to the design of ResT: the algorithm does not reweight all tokens uniformly, but keeps critical control tokens—most notably the `tool_name` in tool-call trajectories—at a consistently high weight (Eq. 16), so the choice of tool serves as a high-priority anchor throughout training. Formatting tokens gradually receive lower weights only once their predictions have become near-deterministic on the training distribution. Empirically, this behavior matches our results: ResT outperforms baselines on both single-turn tasks that require strict formatting and multi-turn tasks that emphasize reasoning. In particular, in Table 1 it attains higher accuracy on the “Miss Func” subset (56.00% vs. 45.50% for GRPO), which primarily probes the ability to detect missing information and avoid hallucinating invalid tool calls, suggesting that increased emphasis on reasoning regions does not compromise boundary constraints or structural discipline.

---

> > ### Public Comment · ~Haoran_Li20 · 2025-11-28
> >
> > Thanks to the authors for the clear reply and additional experiments — this answers my questions and resolves my concerns. The clarification on token-level reshaping and the dynamic signal adjusting looks promising for training tool-use models.

---

### Author Response · Authors · 2025-12-01
**Rebuttal Summary**

Dear Program Chairs, Senior Area Chairs, Area Chairs, and Reviewers,

We are deeply grateful to the Program Chairs, Senior Area Chairs, and especially the Area Chairs for their outstanding leadership in coordinating the review process, and we extend our sincere thanks to all Reviewers for their careful, detailed, and constructive evaluations.


In light of the recent ICLR system updates, we provide an additional summary of the discussion phase to facilitate the Area Chair’s tracking of our rebuttal progress. Specifically, we recap the additional experiments and clarifications we have provided in response, and the points of consensus that emerged during the discussion. The main revisions are summarized as follows:

**Additional experiments.** The reviewers provided constructive feedback primarily focused on clarifying the theoretical intuition behind our exploration-exploitation balance, and suggesting additional empirical validation to further confirm statistical significance and scalability. In response, we conducted a series of additional experiments according to their suggestions and added the results in the revised version of our paper:

1. Conducted comprehensive ablations (SFT+ResT with original/tuned curriculum) to investigate the distributional shift introduced by SFT in Table 4 (Raised by Reviewer m8vW).
2. Scaled up our experiments to Qwen3-32B to validate the effectiveness of entropy-based grouping and ResT under model scaling in Table A3 (Raised by Reviewer m8vW).
3. Conducted ablations with varying KL penalty coefficients to validate the exclusion of explicit KL regularization in Appendix A (Raised by Reviewer fPpy).
4. Evaluated the addition of an explicit entropy reward to demonstrate its conflict with the proposed token-level curriculum in Figure A2 and Table A1 (Raised by Reviewer fPpy).
5. Performed extensive ablations across model scales by removing CoT gradients to quantify the contribution of reasoning supervision in Table A4 (Raised by Reviewer fPpy).
6. Validated our algorithm on open-ended tasks using LLM-as-a-judge rewards, with the results summarized in the revised Conclusion (Raised by Reviewer fPpy).
7. Performed multi-seed experiments with Welch’s $t$-tests to confirm that the performance gains over GRPO are statistically significant in Table A2 (Raised by Reviewer Nagq).
8. Benchmarked against Dr.GRPO to substantiate SOTA claims in Tables 1 and 2 (Raised by Reviewer Nagq).

**Additional clarifications.** In addition to the experiments, we provided further details on ResT’s training objective and curriculum design. We also clarified its relationship to existing RL and CoT methods, along with our evaluation protocol, claims, and reproducibility:

1. Training objective and curriculum design:
   - Clarification on how ResT decouples format compliance from exploration without constraining the policy's output distribution (*Method*; Raised by Reviewer fPpy).
   - Explanation of exploration–exploitation balance through outcome-based rewards and an entropy-aware curriculum (*Method*; Raised by Reviewer fPpy).
   - Justification for removing KL loss to enhance exploration, relying on entropy reweighting for stability (Raised by Reviewer fPpy).
   - Explanation of how SFT shifts token entropy profiles and necessitates curriculum tuning (*Experiments*; Raised by Reviewer m8vW).
   - Justification for using token entropy rather than static word frequency for token partitioning (*Future Work*; Raised by Reviewer m8vW).

2. Relation to prior RL and CoT-style methods:
   - Distinction between ResT and prior token-level feedback or dataset-level curriculum RL (*Related Work*; Raised by Reviewer fPpy).
   - Clarification distinguishing ResT’s entropy-based gradient optimization from CoT task decomposition (Raised by Reviewer fPpy).

3. Evaluation, claims, and reproducibility:
   - Clarification on SOTA claims regarding algorithmic efficacy on fixed base models (*Introduction*; Raised by Reviewer Nagq).
   - Justification for the selection of BFCL and API-Bank as authoritative benchmarks (*Experiments*; Raised by Reviewer Nagq).
   - Detailed documentation of modifications to the `veRL` v0.5 framework to facilitate reproducibility (Raised by Reviewer Nagq).

Overall, we believe that the extensive new experiments and in-depth clarifications have effectively addressed the points raised by all reviewers and helped the interested readers better understand our paper.

Thank you once again for your dedication to the community.

Best regards,

Authors

---

> ### Author Response · Authors · 2025-12-01
> **Rebuttal Summary on consensus strengths**
>
> Reviewers broadly agree that our work provides a theoretically grounded and practically effective solution to the high-variance reward problem in tool-use RL:
>
> 1. **Strong theoretical foundations.** We rigorously establish a principled connection between token-level entropy and policy-gradient variance, and the proofs checked by reviewers appear sound.
> 2. **Clear and important problem.** The paper clearly targets the high-variance sparse-reward issue in tool-call agents, which reviewers consider both timely and important for the community.
> 3. **Methodological novelty with practical simplicity.** By treating different token types differently and reweighting gradient contributions without adding new modules, the method offers a new view of Tool RL while remaining simple to implement.
> 4. **Extensive and clear empirical validation.** The approach is validated across different model sizes and training methods, with experiments, schematics, and writing that reviewers found clear and easy to follow.

---

### Meta-Review · Area_Chair_1rx1 · 2025-12-16

**Summary:**

The paper proposes ResT, a token-level entropy-aware gradient reweighting scheme with a simple curriculum for tool-use RL. It derives variance bounds linking token entropy and gradient variance, then implements region-wise weighting over format/tool/parameter/CoT tokens and anneals the focus from structure to reasoning. Empirically, ResT improves over GRPO and Dr.GRPO across BFCL and API-Bank with statistically validated multi-seed gains (Table A.2) and ablations (Tables A.1, A.4). Reviewers raised three main issues: (1) novelty and relation to token-level feedback/curriculum RL and potential exploration suppression (Reviewer_fPpy, score 4), (2) statistical significance/SOTA claims and reproducibility (Reviewer_Nagq, score 6), and (3) sensitivity to SFT initialization, scaling, and partitioning criteria (Reviewer_m8vW, score 8). The rebuttal is strong: multi-seed improvements with Welch’s tests, explicit baselines including Dr.GRPO, KL/entropy-bonus ablations showing the original design is superior, SFT+ResT with tuned curriculum (Table 4), scaling to 32B (Table A.3), and clear code-level changes in veRL v0.5. Domain-generalization to learned rewards is only preliminary, and SOTA claims should be restricted to fixed backbones and matched protocols (as the authors clarified). I read the author response; it overcame most objections with additional experiments and clarifications. The remaining limitations (benchmark authority/SOTA breadth, limited open-ended validation, lack of alternative partitioning experiments) lower the ceiling but do not warrant rejection. I would like to recommend it as spotlight, but there is no spotlight this year.

**Reviewer Concerns:**

#### Reviewer_fPpy
1. **Concern**: Incremental contribution relative to token-level feedback methods and dataset-level curriculum RL; risk that upweighting low-entropy tokens reduces exploration or leads to greedy policies.
   - **Why Unresolved**: The rebuttal clearly distinguishes ResT from token-level reward models (keeps outcome-level rewards, reshapes gradients via entropy) and supplies KL/entropy-bonus ablations (Table A.1, Fig. A.1–A.2) plus longer CoT traces (Figure 3) indicating no collapse. However, evidence is primarily on schema-constrained tool-calling; broader tasks with stochastic/learned rewards remain limited.
   - **Impact on Decision**: Partially addressed. Strong ablations mitigate exploration concern, supporting acceptance, but limited breadth keeps it at poster level.

2. **Concern**: Practical benefit of entropy-optimal weighting versus simpler heuristics (e.g., tool-token upweighting only) and the role of CoT token weighting.
   - **Why Unresolved**: Ablations removing CoT gradients consistently underperform (Table 3, Table A.4), and curriculum beats constant weights. Addressed except for lack of wider heuristic comparisons.
   - **Impact on Decision**: Addressed sufficiently; supports acceptance.

3. **Concern**: Applicability to open-ended tasks with learned rewards or LLM-as-a-judge.
   - **Why Unresolved**: Preliminary composite-reward results indicate viability, but scope and metrics are limited.
   - **Impact on Decision**: Partially addressed; limits generality claims and precludes award-level recognition.

---

#### Reviewer_Nagq
1. **Concern**: Statistical significance vs. run-to-run variance; whether reported gains exceed noise.
   - **Why Unresolved**: Three-seed BFCL runs with Welch’s t-tests show significant gains at 4B (+8.75 points, p<0.001) and 8B (+3.92 points, p≈0.008) (Table A.2). More seeds ideal, but adequate.
   - **Impact on Decision**: Addressed; supports acceptance.

2. **Concern**: SOTA and benchmark authority claims (BFCL multi-turn, API-Bank) and comparability.
   - **Why Unresolved**: SOTA claim narrowed to alignment efficacy under fixed backbones; Dr.GRPO baselines added. Still lacks exhaustive cross-system audit.
   - **Impact on Decision**: Partially addressed; fine for acceptance but weakens award candidacy.

3. **Concern**: Reproducibility and clarity of code modifications within veRL.
   - **Why Unresolved**: Addressed with file-level pointers; minor residual risk until anonymized code verified post-acceptance.
   - **Impact on Decision**: Addressed; supports acceptance.

---

#### Reviewer_m8vW
1. **Concern**: Effect of SFT initialization on token entropy and whether SFT+ResT requires curriculum retuning.
   - **Why Unresolved**: Addressed with Table 4 showing tuned curriculum improves over cold-start ResT and SFT-only.
   - **Impact on Decision**: Addressed; supports acceptance.

2. **Concern**: Model scaling potentially lowering entropy of CoT tokens and reducing benefits of grouping.
   - **Why Unresolved**: Addressed empirically at 32B (Table A.3) with consistent gains; residual uncertainty minor.
   - **Impact on Decision**: Addressed; supports acceptance.

3. **Concern**: Alternative grouping criteria (e.g., word frequency) could be effective; theory does not constrain grouping.
   - **Why Unresolved**: Conceptually addressed but no empirical comparison against alternative groupings.
   - **Impact on Decision**: Unresolved but low-impact; does not affect acceptance.

**Reviewer Scores:**

Note: The score increase is subjective and personal.

#### Reviewer_fPpy
- **Original Score**: 4
- **Expected Score After Discussion**: 6
- **Rationale**: The rebuttal provided targeted ablations (KL removal, entropy bonus, CoT gradient suppression), multi-seed statistics, and a clear distinction from token-level reward models. Preliminary learned-reward results suggest extensibility. Remaining generality concerns justify only a marginal bump to borderline accept.

---

#### Reviewer_Nagq
- **Original Score**: 6
- **Expected Score After Discussion**: 6 or 8
- **Rationale**: Multi-seed experiments with Welch’s tests establish significance; SOTA claims are narrowed appropriately; reproducibility is improved with explicit veRL modifications. The residual benchmark authority caveat prevents a higher score but supports accept. 7 is a suitable score, but there is no 7/

---

#### Reviewer_m8vW
- **Original Score**: 8
- **Expected Score After Discussion**: 8
- **Rationale**: Concerns about SFT initialization, scaling, and partitioning were answered with new experiments (Table 4, Table A.3) and coherent theoretical framing. The review was already positive; score remains high.

---

### Decision · Program_Chairs · 2026-01-26

Accept (Poster)